# Pan-cancer molecular subtypes revealed by mass-spectrometry-based proteomic characterization of more than 500 human cancers

Fengju Chen[1], Darshan S. Chandrashekar[2,3], Sooryanarayana Varambally[2,3,4] & Chad J. Creighton [1,5,6,7]*

Mass-spectrometry-based proteomic profiling of human cancers has the potential for pan-cancer analyses to identify molecular subtypes and associated pathway features that might be otherwise missed using transcriptomics. Here, we classify 532 cancers, representing six tissue-based types (breast, colon, ovarian, renal, uterine), into ten proteome-based, pan-cancer subtypes that cut across tumor lineages. The proteome-based subtypes are observable in external cancer proteomic datasets surveyed. Gene signatures of oncogenic or metabolic pathways can further distinguish between the subtypes. Two distinct subtypes both involve the immune system, one associated with the adaptive immune response and T-cell activation, and the other associated with the humoral immune response. Two additional subtypes each involve the tumor stroma, one of these including the collagen VI interacting network. Three additional proteome-based subtypes—respectively involving proteins related to Golgi apparatus, hemoglobin complex, and endoplasmic reticulum—were not reflected in previous transcriptomics analyses. A data portal is available at UALCAN website.

[1] Dan L. Duncan Comprehensive Cancer Center Division of Biostatistics, Baylor College of Medicine, Houston, TX, USA. [2] Comprehensive Cancer Center, University of Alabama at Birmingham, Birmingham, AL 35233, USA. [3] Molecular and Cellular Pathology, Department of Pathology, University of Alabama at Birmingham, Birmingham, AL 35233, USA. [4] The Informatics Institute, University of Alabama at Birmingham, Birmingham, AL 35233, USA. [5] Department of Bioinformatics and Computational Biology, The University of Texas MD Anderson Cancer Center, Houston, TX, USA. [6] Human Genome Sequencing Center, Baylor College of Medicine, Houston, TX 77030, USA. [7] Department of Medicine, Baylor College of Medicine, Houston, TX, USA. *email: creighto@bcm.edu

In this age of advanced molecular-profiling technologies, cancer molecular subtype discovery has been one of the more common exercises utilizing transcriptomic or proteomic data on human tumors. Molecular subtypes can deepen our understanding of cancer as representing a collection of diseases rather than a single disease. Molecular subtypes can provide insights into the pathways appearing deregulated within tumor subsets, which may suggest therapeutic opportunities, as well as being indicative of what pathways, as characterized in the experimental setting, would seem particularly relevant in the human disease setting. Historically, most subtype discovery studies in cancer have involved transcriptomic rather than proteomic data, as the advent of DNA microarrays over 20 years ago began the widespread use of transcriptomics among laboratories[1]. In contrast, proteomics is typically more challenging at a technical level and requires dedicated laboratories with the right expertise. In a recent study, using transcriptome data by RNA sequencing (RNA-seq) from The Cancer Genome Atlas (TCGA) consortium, we classified more than 10,000 cancers, representing 32 major types, into 10 molecular-based classes that cut across tumor lineages and cancer types[2]. At the same time, while protein abundance levels typically correlate with those of the corresponding mRNA, widespread discordant expression patterns between protein and mRNA are also observable[3].

The Clinical Proteomic Tumor Analysis Consortium (CPTAC) aims to accelerate the understanding of the molecular basis of cancer through the application of proteomic technologies and workflows to clinical tumor samples[4]. While past TCGA consortium efforts involved targeted proteomics involving a set of ~180–250 protein features[5,6], CPTAC proteomic profiling has been mass-spectrometry-based, profiling on the order of more than 10,000 protein features. Initial studies led by CPTAC performed mass-spectrometry-based profiling on a subset of cases from TCGA—involving breast, colorectal, and ovarian cancers—which allowed for integrative analysis studies between protein expression and other data types including mRNA and mutation[7–9]. Subsequent Confirmatory and Discovery datasets generated by CPTAC profile cancer cases not represented in TCGA[10,11], with these data—representing several tissue-based cancer types—being made publicly available to the research community for secondary analyses.

Mass-spectrometry-based proteomic-profiling data of human cancers, as recently provided by CPTAC for hundreds of cases, have the potential for pan-cancer analyses to identify molecular subtypes and associated pathway features that previous transcriptomics analyses might have missed. Whereas the initial CPTAC-led marker analysis studies were each focused on a specific cancer type[7–11], our present study aims to define pan-cancer proteome-based subtypes that would transcend tissue-based type across the CPTAC cohorts. We also examine cancer proteomic and transcriptomic datasets outside of CPTAC, for patterns of manifestation of our proteome-based subtypes. For each of the subtypes identified, we explore the top differential protein features for associated functional classes and pathways, to help us understand what these subtypes represent.

## Results

**CPTAC mass-spectrometry-based proteomic tumor datasets.** The primary focus of this study was on a set of mass-spectrometry-based proteomic profiles from the CPTAC Confirmatory/Discovery cohort, of 532 cancer cases (cases not represented in TCGA), comprised of 125 breast cancer cases, 97 colon[10], 100 ovarian, 110 renal (primarily clear cell renal cell carcinoma)[11], and 100 uterine (Supplementary Data S1). This CPTAC Confirmatory/Discovery dataset was used to define proteome-based pan-cancer subtypes cutting across tissue-specific differences between the cancer types. Additional proteome and transcriptome datasets of TCGA cancer cases, which shared no cases with the CPTAC Confirmatory/Discovery cohort, were used to independently confirm the observations initially made using CTPAC Confirmatory/Discovery dataset (Fig. 1a). In the CPTAC-TCGA proteome dataset, a subset of 364 cases from TCGA—involving breast, colorectal, and ovarian cancers (105, 90, and 169 cases, respectively, Supplementary Data S1)—was also profiled by mass-spectrometry-based proteomics[7–9]. TCGA RPPA dataset represented 7663 TCGA cases and 31 cancer types profiled for a focused panel of proteins (involving 211 protein

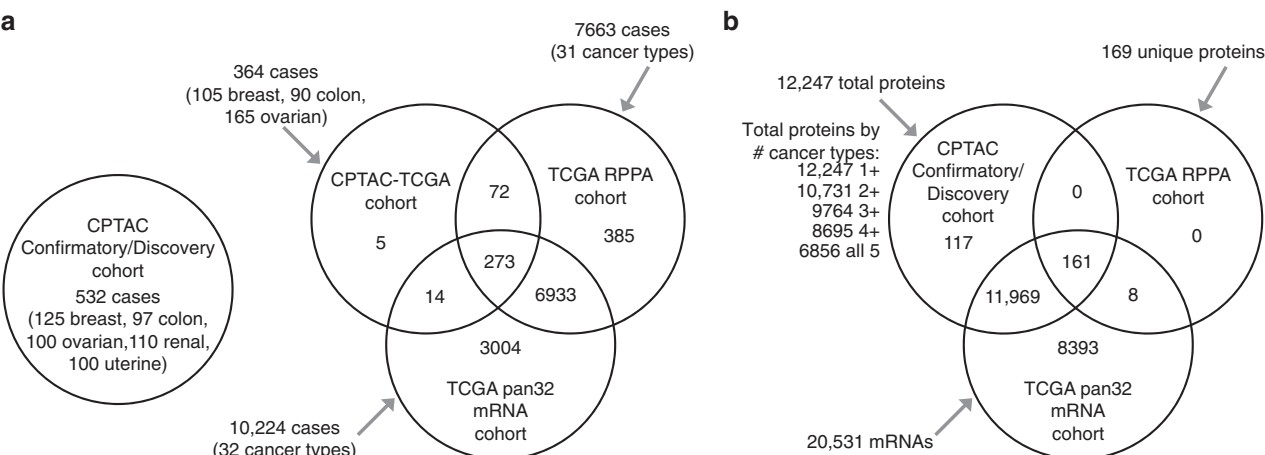

**Fig. 1 Proteomic and transcriptomic human tumor datasets and associated gene features used in this study. a** We used the CPTAC Confirmatory/ Discovery dataset, of 532 cancer cases (cases not represented in TCGA)[10] for proteome-based, pan-cancer subtype discovery (see Fig. 2). For the following datasets of TCGA cases, we classified cases according to the CPTAC Confirmatory/Discovery-based subtypes (see Fig. 3): CPTAC-TCGA dataset (TCGA cases for which mass-spectrometry-based proteomic profiling by CPTAC was carried out), TCGA RPPA dataset (TCGA cases profiled for a focused panel of proteins by RPPA platform), and TCGA pan32 mRNA dataset (TCGA cases with RNA-sequencing data). For TCGA datasets, Venn diagram represents shared cases. **b** Among CPTAC Confirmatory/Discovery, TCGA RPPA, and TCGA pan32 mRNA datasets, numbers of shared gene features (protein or mRNA levels). CPTAC proteomic and TCGA transcriptomic data, as provided by their respective public data portals, were processed at the gene level, rather than at the protein isoform or mRNA transcript levels. See also Supplementary Data S1 and Supplementary Fig. 1.

features[6] or 169 unique genes) by antibody-based reverse-phase protein array (RPPA) platform[12]. TCGA pan32 mRNA dataset featured 10,224 cases and 32 cancer types with RNA-sequencing data. The CPTAC Confirmatory/Discovery total protein dataset represented a total of 12,247 genes, of which 9764 had detection in at least three of the five cancer types (Fig. 1b). For all proteome and transcriptome datasets, we normalized expression values within each cancer type, whereby neither tissue-dominant differences nor inter-laboratory batch effects would drive the downstream analyses[2,5,13].

Previous work had identified 10 molecular-based pan-cancer classes based on transcriptome data from TCGA[2]. Using a previously defined classifier of 854 mRNAs[2], we found these transcriptome-based pan-cancer classes to be reflected in the proteome, based on analysis of both CPTAC-TCGA and CPTAC Confirmatory/Discovery datasets (Supplementary Fig. 1a–c). Within the CPTAC-TCGA cohort, correlations between mRNA and protein expression across tumors were generally positive for the proteins surveyed, though with Pearsons's correlation $r$-values much less than 1 (Supplementary Fig. 1a).

**De novo proteome-based pan-cancer molecular subtypes**. We sought to determine what molecular subtypes would be discoverable from the proteome, as opposed to pan-cancer subtypes previously identified using the transcriptome[2]. We used mass-spectrometry-based proteomic data from the CPTAC Confirmatory/Discovery cohort to define 10 different subtypes of cancer (Table 1 and Supplementary Fig. 2a–d) across the 532 cases and five tissue-based cancer types represented. We found several of these proteome-based subtypes to highly overlap with specific mRNA-based pan-cancer classes[2] as applied to the same cohort (Fig. 2a and Supplementary Fig. 1c). The 10 proteome-based pan-cancer molecular subtypes, referred to here as k1 through k10, were each characterized by widespread molecular patterns (Fig. 2b and Supplementary Fig. 2e, f and Supplementary Data S2 and S3). For each proteome-based subtype, we identified the top proteins most differentially expressed in the given subtype versus the rest of the tumors (Fig. 2b), including both total and phospho-protein features (Supplementary Fig. 3a, b).

We sought to further explore the relationship between our previously defined mRNA-based pan-cancer classes[2] and the proteome-based subtypes of the present study. We assigned each transcriptome profile from TCGA pan32 cohort ($n = 10,224$ cases

and 32 major cancer types) a CPTAC-based pan-cancer subtype. This involved mapping expression values from the top 100 over-expressed proteins for each subtype (Fig. 2b, 1000 protein in total) to the corresponding normalized mRNA values in TCGA dataset. Similar to the above observations in the CPTAC Confirmatory/Discovery cohort (Fig. 2a), several proteome-based subtypes overlapped significantly (one-sided Fisher's exact test) with specific mRNA-based pan-cancer classes (Fig. 2c). In particular, proteome-based versus mRNA-based subtyping[2] associations, respectively, included (proteome-based) k1 to (mRNA-based) c1, k2 to both c3 and c10, k3 to c3, k4 to c5, k5 to c6, k6 to c7, and k7 to c8. The previously identified neuroendocrine-associated c4 class[2] was not well represented among the proteome-based subtypes (Supplementary Fig. 1c). We might attribute this to the most common c4-associated cancer types in TCGA (e.g., cervical, head-and-neck, lung squamous, bladder) not being represented in the CPTAC cohorts. We found each of the proteome-based pan-cancer subtypes to span cases from multiple cancer types (Fig. 2d), with the notable exceptions of k4, which represented the basal-like breast cancer molecular subtype (Supplementary Fig. 4a), and k9, which consisted entirely of clear cell renal cell carcinoma cases. Within the top differentially expressed proteins underscoring each pan-cancer subtype, specific gene categories (by Gene Ontology, or GO, annotation) were over-represented (Fig. 2e and Supplementary Data S4), which associations we further explored below. Differentially expressed proteins within k2, k3, k6, and k7 subtypes in particular represented components of the tumor microenvironment (Supplementary Fig. 4b), which we also further explored below.

In summary, regarding de novo subtype discovery, here we identified 10 pan-cancer subtypes in CPTAC Discovery/Confirmatory cohort, using mass-spectrometry-based proteomics data. To some extent, seven of these ten subtypes are reflected in previously described mRNA-based, pan-cancer subtypes (Table 1).

**Proteome-based subtypes are observable in external datasets**. We examined proteomic and transcriptomic datasets external to the CPTAC Discovery/Confirmatory dataset, for evidence of the manifestation of our proteome-based pan-cancer subtypes. For this we used the top set of 1000 total proteins distinguishing between our 10 subtypes (Fig. 2b) as a classifier (consisting of the top 100 over-expressed proteins per subtype). The subtypes were

**Table 1 Proteome-based pan-cancer molecular subtypes in CPTAC Confirmatory/Discovery cohort.**

| Subtype | n (%) | Associated TCGA mRNA-based class | Description and notable features |
|---|---|---|---|
| k1 | 22 (4.1) | c1 | Over-expression of proteasome complex proteins, glycolysis proteins, and pentose phosphate pathway proteins. |
| k2 | 38 (7.1) | c3,c10 | Adaptive immune system-related; associated with T-cell activation; expression of major histocompatibility complex proteins |
| k3 | 54 (10.2) | c3,c10 | Innate immune system-related; over-expression of complement system proteins; involvement of eosinophils, neutrophils, mast cells, and macrophages; hypoxia signature. |
| k4 | 29 (5.5) | c5 | Represents basal-like breast cancer; over-expression of YAP1 and MYC targets. |
| k5 | 113 (21.2) | c6 | Epithelial signature; normoxia signature; over-expression of YAP1 and MYC targets; over-expression of oxidative phosphorylation and TCA cycle proteins. |
| k6 | 61 (11.5) | c7 | Stromal-related; over-expression of matrix metallopeptidases; Wnt and Notch pathway signatures; hypoxia signature. |
| k7 | 95 (17.9) | c8 | Stromal-related; over-expression of collagen VI proteins; Wnt and Notch pathway signatures. |
| k8 | 43 (8.1) | – | Over-expression of Golgi apparatus-related proteins; Ras pathway signature. |
| k9 | 29 (5.5) | – | Found in clear cell renal cell carcinoma cases only; over-expression of hemoglobin complex proteins. |
| k10 | 48 (9.0) | – | Over-expression of endoplasmic reticulum-related proteins and steroid biosynthesis pathway proteins. |

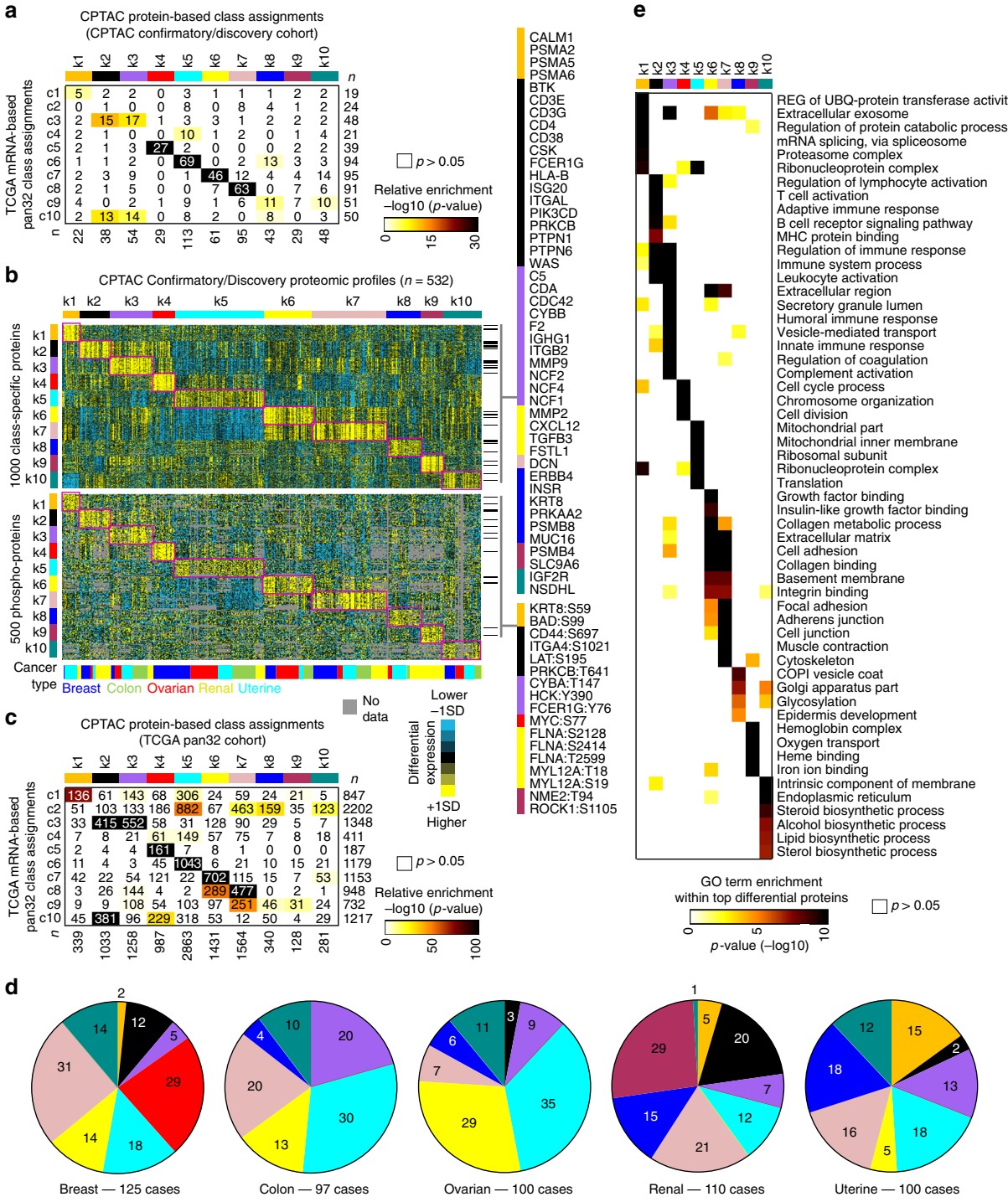

observable in the independent CPTAC-TCGA cohort (Fig. 3a and Supplementary Figs. 4c, d and 5a–c), which we had not used in the subtype discovery. In addition to our classifying the TCGA pan32 mRNA profiles according to proteome-based subtype (Fig. 2c and Supplementary Fig. 6a), we also classified 7694 TCGA cases with RPPA data according to proteome-based pan-cancer subtype, based on protein expression (Fig. 3b). Evidence of the presence of specific proteome-based subtypes in the external cohorts came from observations of subtype calls as made by different data platforms showing significant overlap, e.g., CPTAC-TCGA proteomic subtyping based on mass-spectrometry versus RPPA (Fig. 3c), and TCGA pan32 subtyping based on mRNA versus RPPA (Fig. 3d). Where subtypes did not significantly overlap

between data platforms (e.g. k1 and k9 subtypes), we might attribute this to a number of factors. These factors would include differences of the cohort considered from that of CPTAC Confirmatory/Discovery (e.g., CPTAC-TCGA had no renal cases which were exclusively associated with k9 subtype) and platform-specific differences[12] (e.g., RPPA platform had few available features specific to k8–k10 subtypes, Fig. 3b). We could identify the k4 basal-like breast cancer subtype in external cohorts; still, as the associated proteomic classifier was enriched for cell cycle proteins (Fig. 2e), other tumors that were not breast could also manifest a k4-associated signature pattern. To some degree, several proteome-based subtypes appeared manifested in vitro in cancer cell lines (Supplementary Fig. 6b–d). Permutation testing

**Fig. 2 De novo pan-cancer molecular subtypes as defined by mass-spectrometry-based proteomics. a** By ConsensusClusterPlus[32] of 532 proteomic profiles in the CPTAC Confirmation/Discovery cohort, 10 proteomic-based subtypes—k1 through k10—were identified (columns). For these same cases, pan-cancer class assignments—c1 through c10—based on the previous pan32 mRNA-based discovery[2] were also made (rows, mapping the previous pan32 mRNA classifier to CPTAC protein expression patterns). Significances of overlap between the two sets of classifications are represented. *P*-values by one-sided Fisher's exact test. **b** For CPTAC Confirmation/Discovery cohort, differential expression patterns (values normalized within each tissue-based cancer type; SD, standard deviation from the median) for a set of 1000 proteins (top heat map) and for a set of 500 phospho-proteins (bottom heat map) found to best distinguish between the 10 proteome-based subtypes (see the "Methods" section, top 100 over-expressed proteins for each subtype). Proteins highlighted by name have GO annotation "cell surface receptor signaling pathway" and DrugBank association (lists provide examples of differentially expressed proteins but these would not necessarily have more importance over the other proteins in the heat map, full lists provided in Supplementary Datas 2 and 3). **c** For the TCGA pan32 cohort (*n* = 10,224 cases), we made CPTAC-based pan-cancer subtype assignments (columns, mapping the CPTAC protein expression patterns to TCGA mRNA patterns). Significances of overlap between the CPTAC-based subtypes (columns, k1 through k10) to the previous pan32 mRNA-based pan-cancer class assignments[2] (rows, c1 through c10) are represented. *P*-values by one-sided Fisher's exact test. **d** For each cancer type represented in CPTAC Confirmation/Discovery cohort, distributions by proteome-based subtype. **e** For the top over-expressed proteins associated with each subtype (from part **b**, top panel), represented categories by GO were assessed, with selected enriched categories represented here. *P*-values by one-sided Fisher's exact test. See also Supplementary Figs. 2–4 and Supplementary Data S2 and S3 and S4.

---

demonstrated that the overall strengths of the proteome-based subtype associations in the external datasets were non-random (Supplementary Fig. 7a, b).

In summary, regarding analysis of external profiling datasets of TCGA cases in the context of our proteome-based pan-cancer subtypes, we find most of these subtypes to be observable in TCGA cohort, when considering cases profiled using mass-spectrometry-based proteomics (CPTAC-TCGA), RPPA, or RNA-seq data platforms.

**Pathway-level differences across proteome-based subtypes**. To gain insight into pathways that would distinguish between the various proteome-based pan-cancer subtypes, we applied several pathway-associated gene signatures to CPTAC proteomic expression profiles, as well as to TCGA mRNA expression profiles (Fig. 4a). Many pathways appeared more or less active for different pan-cancer subtypes, as further explored below. Overall, there was broad correspondence in patterns observed between CPTAC proteomic and TCGA mRNA datasets, indicative of associations that would span multiple cohorts and molecular levels, including the associations of hypoxia with k3 and k6 tumors, YAP1 and MYC targets with k4 and k5 tumors, and Wnt and Notch pathways with k6 and k7 tumors. However, there were some notable differences between protein-based and mRNA-based results as well. For example, proteins involved in fatty acid metabolism, glycolysis and gluconeogenesis, pentose phosphate pathway, and tricarboxylic acid (TCA) cycle all were elevated in k1 tumors in CPTAC Confirmatory/Discovery dataset but not in the TCGA pan32 mRNA dataset (Fig. 4a and Supplementary Fig. 8). As another example, oxidative phosphorylation genes appeared elevated at the mRNA level but not at the protein level in k1 tumors. The previously identified c1 pan-cancer class, associated in this study with k1 (Fig. 2c), did, however, show elevation of all of the above metabolic pathways at the mRNA level[2]. As our previous study highlighted the c1, c6, and c8 mRNA-based pan-cancer classes as involving metabolism pathways[2], the k1, k5, and k7 proteome-based subtypes were examined in this present study in a similar manner, by pathway diagram (Fig. 4b). This diagram highlighted both common and distinctive patterns involving individual proteins and mRNAs, including the overall patterns described above.

In summary, regarding associated pathways, we find that each proteome-based subtype is characterized by distinctive pathway-level alterations and enriched functional gene categories. Pathway-level differences include those involving metabolism.

**Immune-related differences across proteome-based subtypes**. The k2 and k3 proteome-based subtypes both involved the

immune system, including protein expression patterns attributable to immune cell infiltrates, with k2 associating with the adaptive immune response and T-cell activation, and with k3 associating with the humoral immune response. The k2 and k3 subtypes were each associated with both the c3 and c10 immune-related mRNA-based classes[2] (Fig. 2a, c). However, while c3 and c10 appeared similar to each other overall, k2 and k3 each involved differential protein expression patterns and associated gene categories that revealed the two subtypes as being quite distinct from each other (Fig. 2b, e). Differentially over-expressed proteins in both k2 and k3 involved GO annotation categories of "regulation of immune response," "immune system process," and "leukocyte activation." Also, k2 but not k3 was enriched for other categories including "T cell activation," "adaptive immune response," and "MHC protein binding"; and k3 was enriched for different categories that included "humoral immune response," "vesicle-mediated transport," and "complement activation."

For CPTAC Confirmatory/Discovery dataset, visualization of the expression patterns for a select set of 162 immune-related proteins, including proteins involving the above GO categories, further demonstrated both common and distinctive expression patterns involving k2 and k3, with a number of the proteins being expressed specifically in immune-related tissues according to Human Protein Atlas[14] (Fig. 5a). Analysis of both gene signatures of infiltrating immune cell types[15] (Fig. 5b) and canonical immune cell markers (Fig. 5c) also revealed differences between k2 and k3. For example, k2 was enriched for signatures and markers of T-cells and antigen-presenting cells, and k3 was enriched for signatures and markers of macrophages, mast cells, eosinophils, and neutrophils. Activation of the complement system in k3 tumors involved both classical and alternative pathways (Fig. 5d). In general, the above associations were also observable in the independent CPTAC-TCGA proteomic and TCGA pan32 mRNA datasets, although k2 and k3 differences were not as distinguishable at the mRNA level (Supplementary Fig. 9a–c).

In summary, regarding immune system-related differences, k2 subtype involved the adaptive immune response and T-cell activation, and k3 subtype involved the humoral immune response. Proteins that distinguish k2 from k3 subtypes include markers of T-cells (high in k2) and markers of mast cells, neutrophils, or macrophages (all high in k3), as well as complement system pathway proteins (high in k3). These distinctions between k2 and k3 were not evident in previous mRNA-based subtyping[2], where k2 and k3 tumors associated together as a single group.

**Stroma-related differences across proteome-based subtypes**. The k6 and k7 proteome-based subtypes both involved the tumor

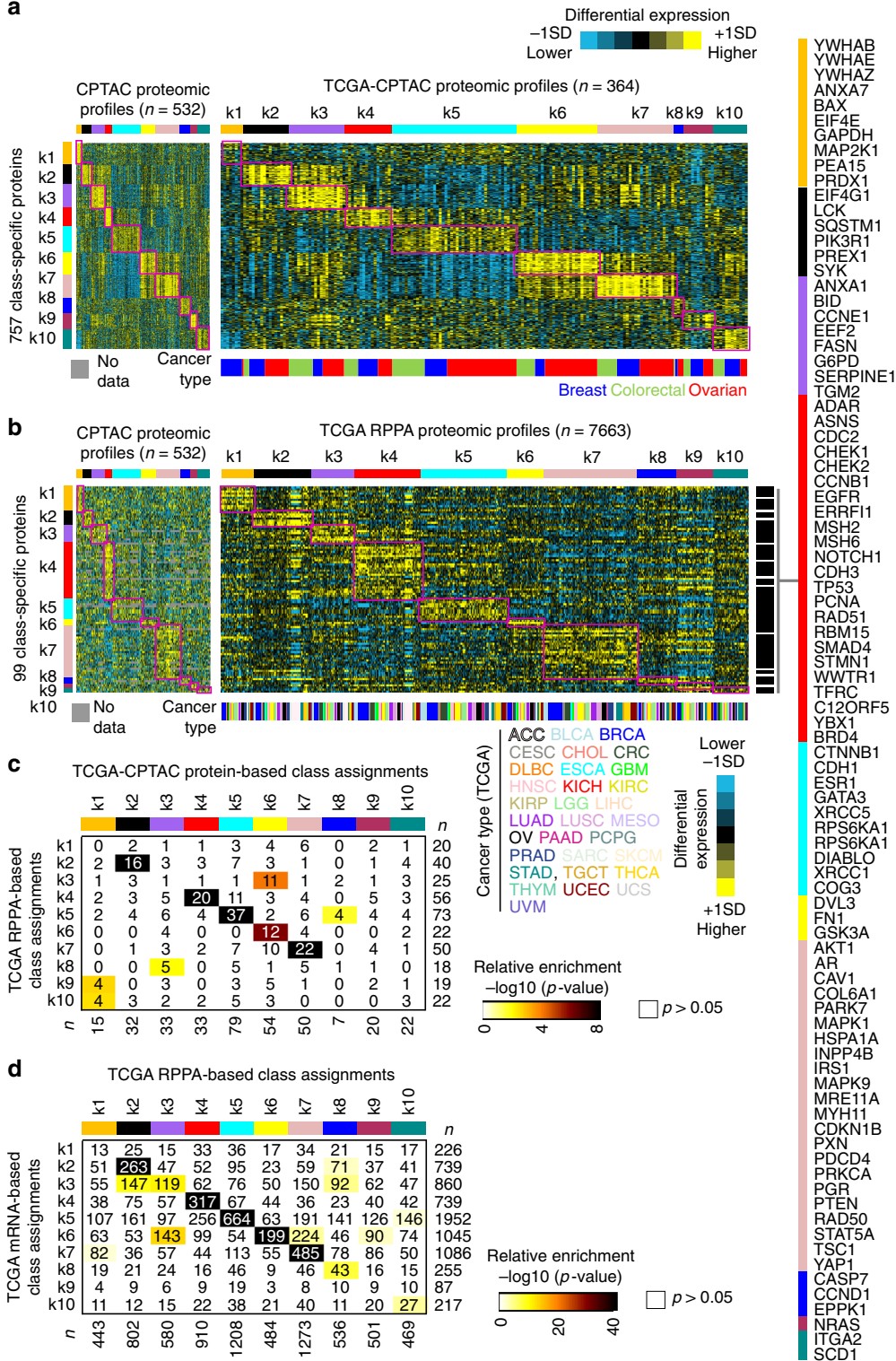

stroma, including protein expression patterns likely attributable to non-cancer cells and the tumor microenvironment[2]. The k6 and k7 subtypes were, respectively, associated with the c7 and c8 stroma-related mRNA-based classes[2] (Fig. 2a, c), further reinforcing the notion of the distinctions between these subtypes representing various biological roles for the stromal component in human cancer[2]. Differentially over-expressed proteins in both k6 and k7 involved GO annotation categories (Fig. 2e) of "extracellular matrix," "cell adhesion," "collagen binding," and "basement membrane." Also, k6 but not k7 was enriched for

other categories including "growth factor binding," and k7 was enriched for different categories that included "muscle contraction" and "cytoskeleton." For CPTAC Confirmatory/Discovery dataset, visualization of the expression patterns for a select set of 606 extracellular matrix-related proteins, including proteins involving the above GO categories, further demonstrated both common and distinctive expression patterns involving k6 and k7 (Fig. 6a). In general, associations involving the 606 proteins were also observable in the independent CPTAC-TCGA proteomic and TCGA pan32 mRNA datasets, although the differences were

**Fig. 3 Observation of CPTAC pan-cancer proteome-based subtypes in additional multi-cancer protein expression profiling datasets. a** The 364 TCGA cases with mass-spectrometry-based proteomic data from CPTAC were classified according to proteome-based pan-cancer subtype as originally defined using CPTAC Confirmatory/Discovery cohort. Expression patterns for the top set of 757 proteins distinguishing between the 10 subtypes (from Fig. 2a, based on available data) are shown for both CPTAC Confirmatory/Discovery and CPTAC-TCGA proteomic datasets (values normalized within each tissue-based cancer type; SD, standard deviation from the median). Gene patterns in the CPTAC-TCGA sample profiles sharing similarity with a subtype-specific signature pattern are highlighted. **b** The 7694 TCGA cases with reverse-phase protein array (RPPA) data were classified according to proteome-based pan-cancer subtype. Expression patterns for a top set of 99 proteins distinguishing between the 10 subtypes (see the section "Methods", based on available data) are shown for both CPTAC Confirmatory/Discovery and TCGA RPPA proteomic datasets. Gene patterns in the RPPA sample profiles sharing similarity with a subtype-specific signature pattern are highlighted. Proteins highlighted by name were individually significantly associated with the given subtype ($P < 0.001$, $t$-test) in TCGA RPPA dataset. **c** Significances of overlap between the proteome-based subtype assignments made for the CPTAC-TCGA dataset (columns), with proteome-based subtype assignments for the TCGA RPPA dataset (rows), based on the 345 cases represented in both datasets. $P$-values by one-sided Fisher's exact test. **d** Significances of overlap between the proteome-based subtype assignments made for the TCGA RPPA dataset (columns), with subtype assignments for the transcriptome profiles in TCGA pan32 cohort (rows, mapping the CPTAC protein expression patterns to TCGA mRNA patterns), based on the 7206 cases represented in both datasets. $P$-values by one-sided Fisher's exact test. Patient-level subtyping and cancer type information for all datasets represented are provided in Supplementary Data 1. See also Supplementary Figs. 5–7.

not as distinguishable at the mRNA level (Supplementary Fig. 10). Also, k6 tumors showed under-expression for proteins related to the TCA cycle and oxidative phosphorylation (Fig. 4a and Supplementary Fig. 8) and over-expression of matrix metallopeptidases including MMP11, MMP13, and MMP14 (Fig. 6a). Integration of the set of proteins high in either k6 or k7 tumors with public databases of protein–protein interactions generated protein interaction networks (Fig. 6b and Supplementary Data S5), which allowed visualization of the potential relationships involving these proteins. While k6 and k7 over-expressed many of the same proteins, one feature distinguishing k7 from k6 was over-expression of collagen VI members, with many collagen VI interacting proteins[16] also being preferentially higher in k7 tumors (Fig. 6c).

In summary, regarding tumor stroma-related differences, k6 and k7 subtypes showed both common and distinctive expression patterns with respect to each other, involving the influence of the tumor microenvironment. Protein markers that distinguish k6 from k7 subtypes include: FN1, IGFBP3, ITGAV, LOX, LOXL2, MMP11, MMP13, MMP14, and THBS1 (all high in k6, Fig. 6a); and collagen VI and associated proteins (high in k7, Fig. 6c). The distinctions between k6 and k7 are reflected in profiling datasets external to CPTAC, including RNA-seq-profiling datasets.

**Proteome-based subtypes not reflected in the transcriptome.** Interestingly, the k8, k9, and k10 proteome-based subtypes as discovered in the CPTAC Confirmation/Discovery cohort were not well-represented in the other proteomic and transcriptomic datasets examined (Fig. 2c, d and Supplementary Fig. 6d). For k8 and k10, however, there was some significant overlap ($p < 1E-6$ and $p < 0.001$, respectively, one-sided Fisher's exact test) between RPPA-based and mRNA-based subtype calling in TCGA pan32 cohort (Fig. 3d). The k9 subtype consisted entirely of clear cell renal cell carcinoma cases, which cancer type was not represented in the CPTAC-TCGA cohort. Besides, the RPPA platform had few available features specific to k8, k9, and k10 subtypes (Fig. 3a), which could complicate assigning proteome-based subtype to RPPA profiles. Nevertheless, significant numbers of top differentially expressed proteins—with associated GO categories being over-represented—underscored each of the above subtypes (Fig. 1e, f), indicative of the subtypes representing actual biology. Proteins high in k8 included Golgi-apparatus-related proteins (Figs. 1e and 7a), proteins high in k9 included hemoglobin complex proteins (Figs. 1e and 7b), and proteins high in k10 included endoplasmic reticulum (ER)-related proteins and proteins involved in steroid biosynthesis (Figs. 1e, 7c, d). Typically, sterols are synthesized in the ER and transported by non-vesicular mechanisms to the plasma membrane[17]. Ras pathway

signature also appeared manifested in k8 tumors (Fig. 4a). Integration of the set of proteins high in k8, k9, or k10 tumors with public databases of protein–protein interactions generated protein interaction networks (Fig. 7e), where a number of the interacting proteins showed concordant patterns in the CPTAC-TCGA proteomic and TCGA pan32 mRNA datasets.

In summary, regarding the proteome-specific subtypes, k8 involved Golgi apparatus-related proteins, k9 (specific to renal cases) involved hemoglobin complex proteins, and k10 involved ER-related proteins and steroid biosynthesis pathway proteins.

**A portal for visualizing proteomic associations by subtype.** To facilitate access to CPTAC proteomic results by the general biomedical research community, we have integrated CPTAC data with the UALCAN data portal[18]. Across tumor and normal samples in the CPTAC Confirmatory/Discovery datasets, the UALCAN interface (http://ualcan.path.uab.edu) allows the user to analyze relative expression levels of a query protein or set of proteins across specified tumor sub-groups. These pre-defined tumor sub-groups may include cancer stage, tumor grade, race, or other clinicopathologic features. Proteome-based subtype comparisons (k1–k10), either across all five cancer types surveyed ("pan-cancer" view) or within a given cancer type, can also be carried out for a given protein. The analysis results (e.g., box plots) can be printed directly or downloaded in several file formats compatible with presentations or figures for publication. Users can also examine TCGA datasets in UALCAN, for differential patterns involving mRNA or DNA methylation features corresponding to the proteins of particular interest arising from the analysis of CPTAC data.

**Discussion**
Our proteome-based, pan-cancer subtypes represent another view into the molecular landscape of cancer, distinct in many ways from previous transcriptome-centered views. These proteome-based subtypes provide a framework for examining pathways or processes that, at the protein level, would cut across individual cancer types. The molecular landscape of human cancer can help guide us as to what pathways that have been extensively studied in the experimental setting would be particularly relevant in the setting of human disease. In this present study, many such pathways are found to be differentially expressed or manifested within specific subtypes of human cancers. Other pathways not examined here may also be explored in the context of our proteome-based subtypes. To this end, we have added the CPTAC datasets to the UALCAN data portal[18], as well as provided protein-level statistics for all proteins in the supplementary data of this study, allowing researchers to look up proteins of

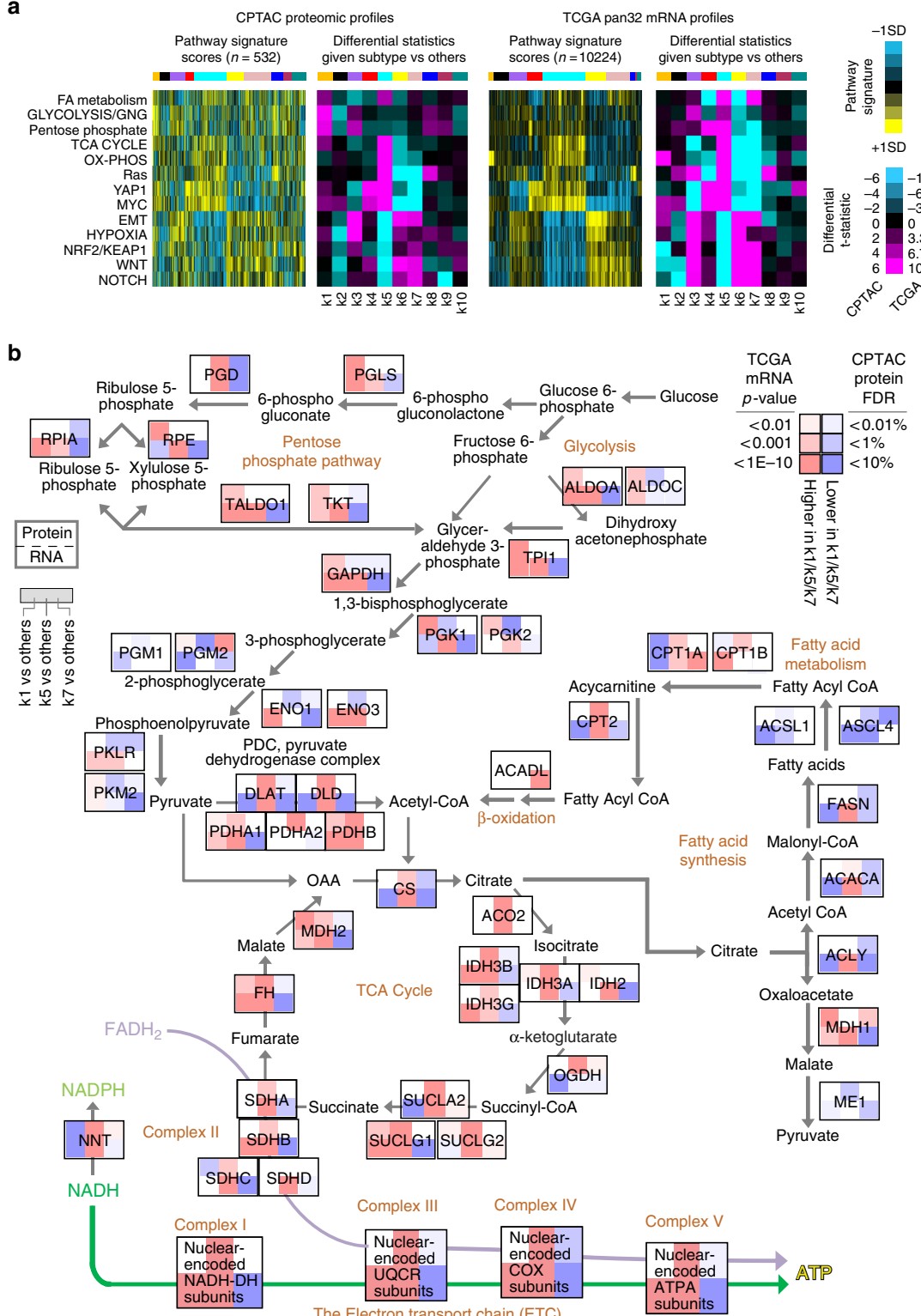

**Fig. 4 Proteome-based subtype-specific differences involving metabolic pathways. a** For CPTAC Confirmatory/Discovery proteomic dataset and for TCGA pan32 mRNA dataset, pathway-associated gene signatures (using values normalized within each cancer type; SD, standard deviation from the median). For each dataset, purple-cyan heat maps denote *t*-statistics for comparing the given subtype versus the other tumors (bright purple/cyan, highly significant; black, not significant; shades close to black, borderline significant). Selected pathways surveyed by signatures[2] included several related to metabolism (FA fatty acid; GNG gluconeogenesis; TCA tricarboxylic acid; OX-PHOS oxidative phosphorylation). **b** Pathway diagram representing core metabolic pathways, with differential expression patterns represented (using values normalized within cancer type), comparing tumors in pan-cancer subtypes k1, k5, or k7 with the rest of the tumors. For each protein represented, the top portion represents results from differential protein analysis (CPTAC Confirmatory/Discovery proteomic dataset) and the bottom portion represents results from differential mRNA analysis (TCGA pan32 mRNA dataset). Red denotes significantly higher expression in k1/k6/k8 and blue denotes significantly lower expression. See also Supplementary Fig. 8.

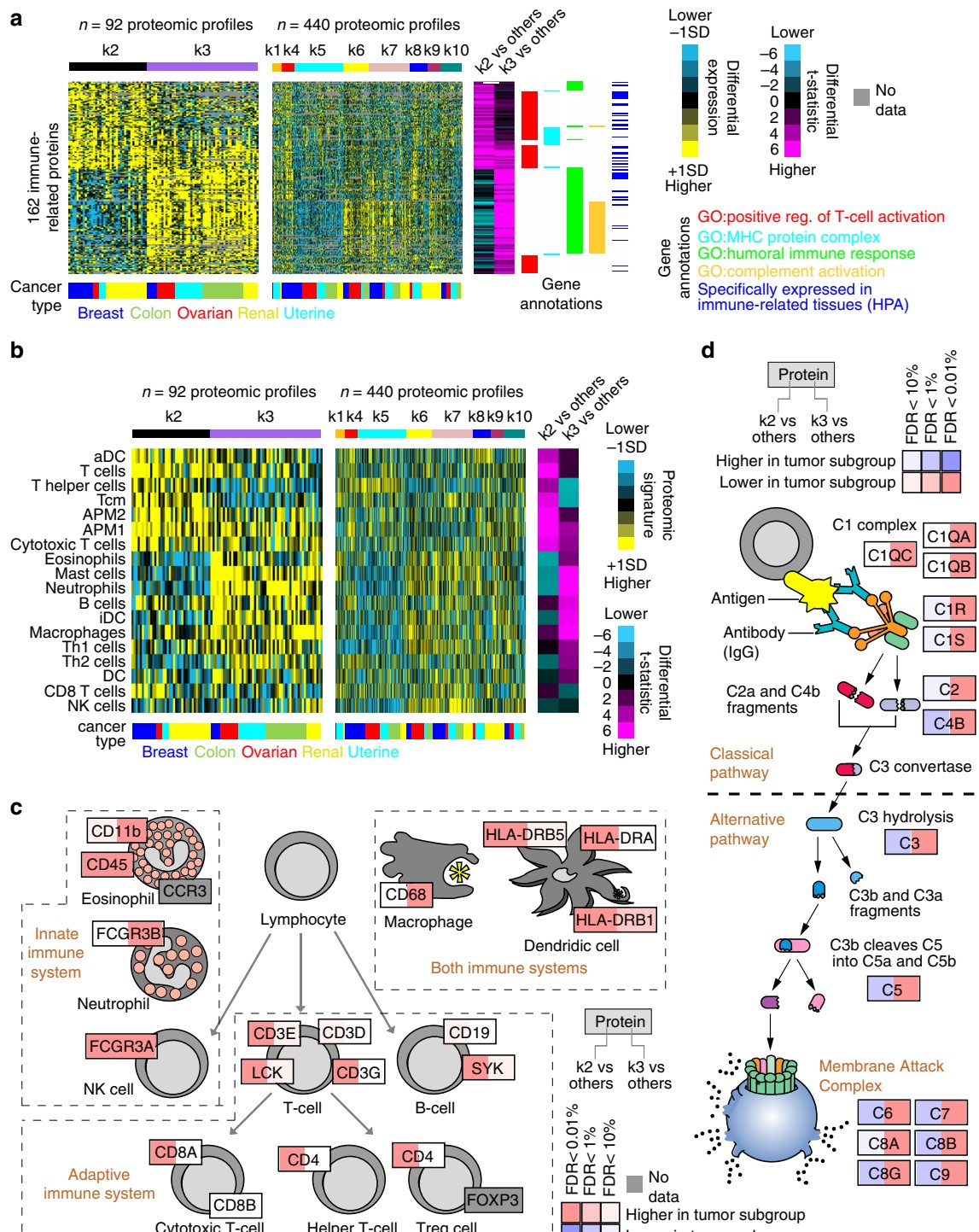

**Fig. 5 Immune system-related differences underscore k2 and k3 proteome-based subtypes. a** For a set of 162 immune-related proteins (FDR < 5% for either k2 or k3 subtypes and association with one of the indicated GO annotation categories), heat maps of differential protein expression patterns (expression values normalized within cancer type; SD, standard deviation from the median), across CPTAC Confirmatory/Discovery proteomic profiles, ordered by subtype. Purple-cyan heat map denotes *t*-statistics for comparing the given subtype versus the other tumors (bright purple/cyan, highly significant; black, not significant; shades close to black, borderline significant). Proteins found specifically expressed in immune-related tissues (according to Human Protein Atlas[14], or HPA, www.proteinatlas.org) are indicated. **b** Heat maps of gene expression-based signatures[15] of immune cell infiltrates, across CPTAC Confirmatory/Discovery proteomic profiles, ordered by subtype (expression values normalized within cancer type; SD, standard deviation from the median). Purple-cyan heat map denotes *t*-statistics for comparing the given subtype versus the other tumors. APM1/APM2, antigen presentation on MHC class I/class II, respectively; DC dendritic cells; iDC immature DCs; aDC activated DCs; NK cells natural killer cells; Tcm cells T central memory cells; Tem cells T effector memory cells. **c** Diagram of immune cell types and associated protein markers. Red denotes significantly higher expression in k2 or k3 subtypes as indicated, and blue denotes significantly lower expression. FDR false discovery rate. **d** Similar to part **c**, but for complement activation pathway. See also Supplementary Fig. 9.

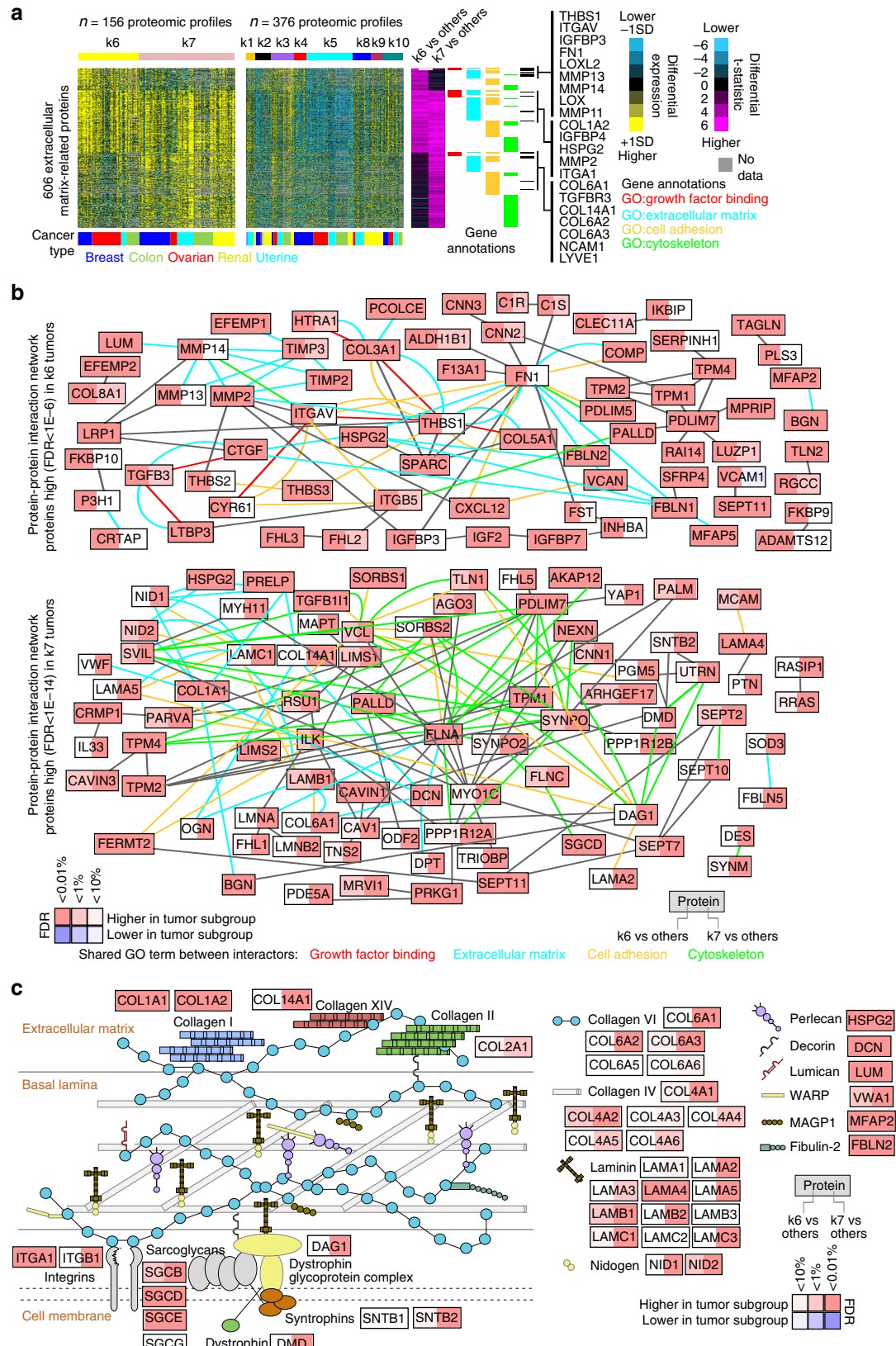

individual interest. In this way, our study and the associated data provide an excellent resource for future studies.

Four of the 10 proteome-based subtypes represent the involvement of non-cancer cells, two subtypes (k2 and k3) involving immune cells and the other two subtypes (k6 and k7) involving the tumor stroma or reactive stroma[19]. While it is understood

that tumor sample purity factors into the global expression profile[20], the above four subtypes all appear very distinct from each other. This indicates that the associated non-cancer proteomic patterns represent true biology rather than a purely technical artifact involving sample procurement. Also, multiple processes that we would associate with the tumor microenvironment are

**Fig. 6 Tumor stroma-related differences underscore k6 and k7 proteome-based subtypes. a** For a set of 606 extracellular matrix-related proteins (FDR < 5% for either k6 or k7 subtypes and association with one of the indicated GO annotation categories), heat maps of differential protein expression patterns (expression values normalized within cancer type; SD, standard deviation from the median), across CPTAC Confirmatory/Discovery proteomic profiles, ordered by subtype. Purple-cyan heat map denotes *t*-statistics for comparing the given subtype versus the other tumors (bright purple/cyan, highly significant; black, not significant; shades close to black, borderline significant). Selected proteins of interest are listed by name. **b** Protein–protein interaction networks involving the top proteins over-expressed in k6 tumors (top network, using cutoff of FDR < 1E−6) and the top proteins over-expressed in k7 tumors (bottom network, using cutoff of FDR < 1E−14). Nodes represent proteins that were found over-expressed in either k6 or k7 subtypes as indicated. Red node coloring denotes significantly higher expression in k6 or k7 subtypes as indicated, and blue coloring denotes significantly lower expression. A line between two nodes signifies that the corresponding protein products of the genes can physically interact (according to the literature, from Entrez gene interactions database). Colored edges (other than gray) denote a common GO term annotation shared by both of the connected proteins. **c** Diagram of collagen VI interactions and associated proteins[16]. Red denotes significantly higher expression in k6 or k7 subtypes as indicated, and blue denotes significantly lower expression. FDR false discovery rate. See also Supplementary Fig. 10 and Supplementary Data S5.

manifested within distinct subtypes of tumors, indicating that these processes are acting independently from each other. While the recent surge of interest in cancer immunotherapy is mainly focused on T cell function or numbers, which cells appear most present within our k2 subtype, relatively less attention has been placed on the complement activation pathway, represented in our k3 subtype. Activation of the complement system is an important component of tumor-promoting inflammation, which in turn has an important role in carcinogenesis and cancer progression[21], and which involves macrophages and neutrophils. Notably, hypoxia was found here to be elevated in the k3 subtype, while elsewhere hypoxia is known to alter the regulation of complement proteins in different cellular components of the tumor microenvironment[22]. Previous surveys of TCGA data for immune cell involvement at the transcriptomic level[2,23] appear to have missed the complement pathway as representing an important player. On the other hand, collagens, which appear to have a role in our k6 and k7 subtypes, constitute the scaffold of the tumor microenvironment. Collagens affect this microenvironment such that it regulates extracellular matrix remodeling by collagen degradation and re-deposition, and promotes tumor infiltration, angiogenesis, invasion, and migration[24].

Three additional proteome-based subtypes—k8, k9, and k10—were not reflected in previous pan-cancer transcriptomics analyses[2] and relatively few of the >10,000 cases in TCGA could be assigned to them based on gene transcription, suggesting that these subtypes might be unique to the proteome. The classes of proteins associated with these subtypes may suggest pathways or processes that may not be as commonly associated with human cancer, but for which links to cancer can be found in the literature. Regarding the k8 subtype, many studies have demonstrated the essential roles of the Golgi in cellular activities as a stress sensor, apoptosis trigger, lipid/protein modifier, mitotic checkpoint, and a mediator of malignant transformation[25]. Regarding the k9 subtype, specific to renal cell carcinoma in the CPTAC cohort, increased hemoglobin has been associated elsewhere with VEGF inhibitor treatment in advanced renal cell carcinoma[26], and elevation in hemoglobin on VEGF-directed therapy has been associated with worse clinical outcomes[27]. Regarding the k10 subtype, tumor cells are often exposed to intrinsic and external factors that alter protein homeostasis, thus producing ER stress. ER stress, in turn, may be activated by a variety of factors and triggers the unfolded protein response (UPR), which restores homeostasis or activates cell death[28,29]. In the future, as greater numbers of human cancer cases are profiled using mass-spectrometry-based proteomics, additional subtypes and associated pathways may be uncovered and explored. Such findings could represent key biological insights and potential therapeutic opportunities involving appreciable subsets of human cancer.

In summary, our study has uncovered proteome-based pan-cancer subtypes on the basis of mass-spectrometry-based proteomics, which platform offers far more protein features over

other proteomic platforms such as RPPA[5]. Where specific proteome-based subtypes are found reflected in subtypes defined on the basis of mRNA patterns[2], the existence of such subtypes in human cancer has even stronger support with these independent observations involving the proteome in addition to the transcriptome. On the other hand, our study also uncovers cancer subtypes not found in previous transcriptome-based studies. Observations specific to the proteome include two different immune system-related subtypes (reflected as a common subtype in the transcriptome), one involving the adaptive immune response and the other involving the humoral immune response. The pathway associations according to proteome-based subtype may help in drawing attention to pathways and processes, previously examined in the laboratory setting, which are shown here to be involved in appreciable subsets of human cancer. In addition to the more heavily studied pathways, such as cancer metabolism and T cell signaling, other pathways and processes to be considered further in the context of cancer would include those involving complement activation, collagen VI, Golgi apparatus, hemoglobin complex, and ER.

## Methods

**CPTAC datasets**. The National Cancer Institute CPTAC[30] generated the mass spectrometry-based proteomic data used in this publication. The CPTAC Confirmation/Discovery cohort, used to define proteome-based molecular subtypes, consisted of five separate datasets: CPTAC Uterine Corpus Endometrial Carcinoma (UCEC) Discovery Study (comprising *n* = 100 cases), CPTAC Clear Cell Renal Cell Carcinoma (CCRCC) Discovery Study (*n* = 110)[11], CPTAC Breast Cancer Confirmatory Study (*n* = 125), CPTAC Ovarian Cancer Confirmatory Study (*n* = 100), and CPTAC Colon Cancer Confirmatory Study (*n* = 97)[10]. The CPTAC-TCGA datasets, used here for independent observations or validations, involved 364 cancer cases in TCGA—including 90 colorectal[9] cases, 105 breast[8] cases, and 169 ovarian[7] cases. The mass-spectrometry-based proteomics methods for profiling CPTAC-TCGA tumors have been previously reported in the associated CPTAC-led studies, as well as broadly summarized below. Tumor samples from CCRCC and UCEC were analyzed by global proteomic and phosphoproteomic mass spectrometry using the 10-plexed isobaric tandem mass tags (TMT-10), following the CPTAC reproducible workflow protocol published by Mertins et al.[31]. Breast, colon, and ovarian samples were analyzed with liquid chromatography–tandem mass spectrometry (LC–MS/MS) global proteomic and phosphoproteomic profiling. Proteomic-profiling data were generated through informed consent as part of CPTAC efforts and analyzed per CPTAC data use guidelines and restrictions. For CPTAC Confirmatory/Discovery, we obtained processed protein expression data from the CPTAC Data Portal[4]. For CPTAC-TCGA, we obtained processed protein expression data from the supplementary tables of the associated publications. CPTAC proteomic data, as provided by the CPTAC Data Portal and related publications, were processed at the gene level, rather than at the protein isoform level; as a simplification, we did not consider different isoforms for the same protein in the present study.

Taking the expression values provided in the Protein Report provided by CPTAC Data Portal, we normalized CPTAC Confirmatory/Discovery proteomic data for downstream analyses in the following manner. First, within each proteomic profile, we normalized logged expression values to standard deviations from the median. We carried out the above as, for some cancer types, there was an observed high variability in within-profile expression median or standard deviation across samples, which could influence unsupervised analysis results. Next, we normalized expression values across samples to standard deviations from the median. In the same manner, we separately normalized both total protein and phospho-protein datasets for a given cancer type. For the Colon Confirmatory

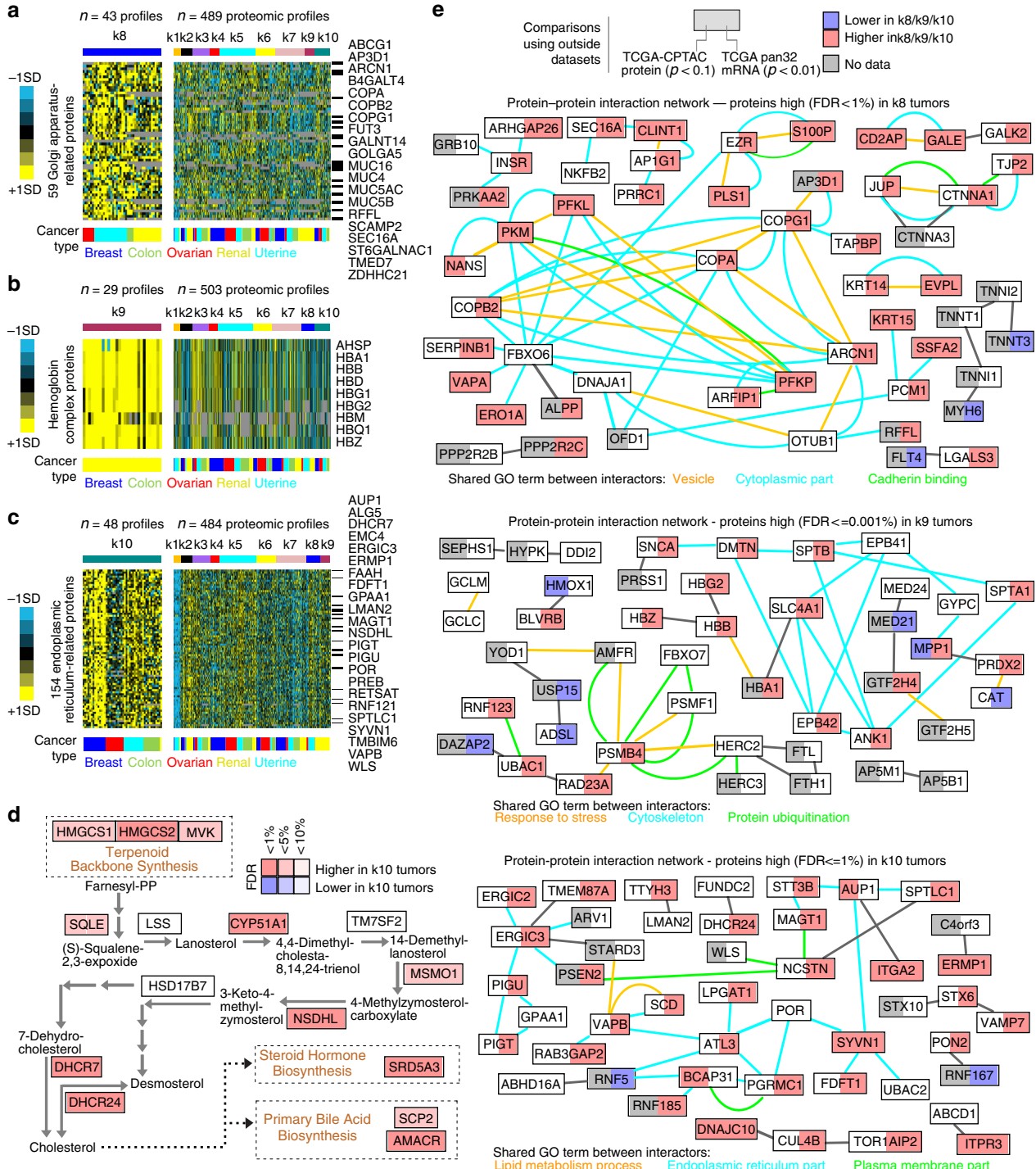

**Fig. 7 Overview of k8, k9, and k10 proteome-based subtypes. a** For a set of 59 Golgi apparatus-related proteins elevated in the k8 subtype (FDR < 5%), heat maps of differential protein expression patterns (expression values normalized within cancer type; SD, standard deviation from the median), across CPTAC Confirmatory/Discovery proteomic profiles ordered by subtype. Listed proteins are the subset highest in k8 over all other subtypes. **b** For a set of nine hemoglobin complex proteins elevated in the k9 subtype (FDR < 1%), heat maps of differential protein expression patterns, across CPTAC Confirmatory/Discovery proteomic profiles ordered by subtype. **c** For a set of 154 endoplasmic reticulum-related proteins elevated in the k10 subtype (FDR < 1%), heat maps of differential protein expression patterns, across CPTAC Confirmatory/Discovery proteomic profiles ordered by subtype. Listed proteins are those among the top 50 most over-expressed for the k10 subtype. **d** Diagram of steroid biosynthesis pathway and associated proteins (from KEGG database[48]). Red denotes significantly higher expression in the k10 subtype. **e** Protein–protein interaction networks involving the top proteins over-expressed in k8 tumors (top network), the top proteins over-expressed in k9 tumors (middle network), and the top proteins over-expressed in k10 tumors (bottom network). Nodes represent proteins that were found over-expressed in the given subtype. Nodes are colored according to patterns of differential expression in additional cohorts (left, protein data from CPTAC-TCGA cohort; right, mRNA data from TCGA pan32 cohort). A line between two nodes signifies that the corresponding protein products of the genes can physically interact (according to the literature, from Entrez gene interactions database). Colored edges (other than gray) denote a common GO term annotation shared by both of the connected proteins. FDR false discovery rate.

dataset, we used the data from Pacific Northwest National Laboratory (PNNL) in the analysis. For the Ovarian Confirmatory dataset, we averaged normalized values for PNNL and Johns Hopkins University (JHU) in instances where there were duplicate profiles for the same sample. As intended, by normalizing expression within each cancer type and within each CPTAC dataset, neither tissue-dominant differences nor inter-laboratory batch effects would drive the downstream unsupervised subtype discovery.

For the CPTAC Confirmatory/Discovery total protein dataset, a total of 12,247 unique genes by Entrez Identifier comprised the compiled dataset of all five cancer types. For unsupervised subtype discovery and selecting top protein features for the subtype classifier, we considered for this analysis the subset of 9764 unique proteins represented in at least three of the five cancer types. For the CPTAC Confirmatory/ Discovery phospho-protein dataset, we considered for this analysis the set of protein features detected in more than half of cases for at least three cancer types. For the CPTAC-TCGA total protein dataset, we considered for this analysis the set of 5863 unique genes/proteins by Entrez identifier, which proteins were represented in all three cancer types.

**TCGA datasets.** Results are based in part upon data generated by TCGA Research Network (http://cancergenome.nih.gov/). We aggregated TCGA transcriptomic and RPPA data from public repositories, listed in the "Data availability" section. RNA-seq expression data were processed by TCGA at the gene level, rather than at the transcript level. Tumors spanned 32 different TCGA projects, each project representing a specific cancer type, listed as follows: LAML, acute myeloid leukemia; ACC, adrenocortical carcinoma; BLCA, bladder urothelial carcinoma; LGG, lower grade glioma; BRCA, breast invasive carcinoma; CESC, cervical squamous cell carcinoma and endocervical adenocarcinoma; CHOL, cholangiocarcinoma; CRC, colorectal adenocarcinoma (combining COAD and READ projects); ESCA, esophageal carcinoma; GBM, glioblastoma multiforme; HNSC, head and neck squamous cell carcinoma; KICH, kidney chromophobe; KIRC, kidney renal clear cell carcinoma; KIRP, kidney renal papillary cell carcinoma; LIHC, liver hepatocellular carcinoma; LUAD, lung adenocarcinoma; LUSC, lung squamous cell carcinoma; DLBC, lymphoid neoplasm diffuse large B-cell lymphoma; MESO, mesothelioma; OV, ovarian serous cystadenocarcinoma; PAAD, pancreatic adenocarcinoma; PCPG, pheochromocytoma and paraganglioma; PRAD, prostate adenocarcinoma; SARC, sarcoma; SKCM, skin cutaneous melanoma; STAD, stomach adenocarcinoma; TGCT, testicular germ cell tumors; THYM, thymoma; THCA, thyroid carcinoma; UCS, uterine carcinosarcoma; UCEC, uterine corpus endometrial carcinoma; UVM, uveal melanoma. Cancer molecular profiling data were generated through informed consent as part of previously published studies and analyzed per each original study's data use guidelines and restrictions.

**Pan-cancer molecular subtype discovery.** ConsensusClusterPlus R-package[32] was used to identify the structure and relationship of the samples. For unsupervised clustering analysis, we selected the top 2000 most variable proteins from the CPTAC Confirmatory/Discovery total protein dataset (taken from the set of 9764 unique proteins represented in at least three of the five cancer types), according to average standard deviation (using log-transformed expression values centered to standard deviations from the median within each cancer type) across the five CPTAC projects. Consensus ward linkage hierarchical clustering identified $k = 2$ to $k = 15$ subtypes, with the stability of the clustering increasing with increasing $k$. Consistent with what was carried out for our previous studies[2], we selected the $k = 10$ clustering solution for further investigation. By additional clustering solutions more subtypes may potentially be discoverable, although these additional subtypes would involve fewer numbers of cases and would represent less of the global variation (Supplementary Fig. 2).

**Analysis of external multi-cancer datasets.** We examined external, multi-cancer gene expression profiling datasets, classifying each external tumor profile by pan-cancer class/subtype as defined by TCGA or CPTAC data. Within each cancer type of the external dataset being classified, we normalized log-transformed genes or proteins to standard deviations from the median. As a classifier, we used either the top set of mRNAs (from TCGA-based pan32 study, Fig. 1a and ref. [2]) or the top set of proteins (from the present study, Fig. 2) distinguishing between the pan-cancer subtypes, as noted. For each pan-cancer subtype, we computed the average normalized value for each gene or protein, based on the centered expression data matrix. We then computed the Pearson's correlation between each external profile and each pan-cancer subtype averaged profile. We assigned each external cancer case to a pan-cancer subtype, based on which subtype profile showed the highest correlation with the given external dataset profile. Supplementary Data 3 provides example calculations in Excel, by which the CPTAC-TCGA proteomic profiles are classified according to proteome-based pan-cancer subtype. We re-classified the CPTAC-TCGA mRNA profiles according to TCGA mRNA-based molecular class, with the same set of 198 class-specific features shared between protein and mRNA datasets being used; we assigned profiles for which the best class fit had a Pearson's correlation of >0.05 to the c2 class (the class lacking strongly associated expression patterns according to the original Chen et al. [2] study).

By the above approach, the CPTAC total protein datasets (both CPTAC-TCGA and CPTAC Confirmatory/Discovery) were separately classified according to TCGA

mRNA-based pan-cancer class (Fig. 1). The previously defined mRNA-based classifier consisted of 854 genes, of which 198 were represented in the CPTAC-TCGA dataset (taking from the set of 5863 unique genes/proteins represented in all three cancer types), and 532 were represented in CPTAC Confirmatory/Discovery dataset (taking from the set of 12,247 unique genes represented in any one of the five cancer types). We classified TCGA pan32 mRNA profiles ($n = 10,224$ cases) according to proteome-based subtype as derived using the CPTAC Confirmatory/ Discovery dataset (Fig. 2c and Supplementary Fig. 6a, using 990 out of 1000 features in Fig. 2b classifier with available data), mapping protein features to mRNA by Entrez gene identifier. We classified the CPTAC-TCGA proteomic profiles according to the subtype originally derived from the CPTAC Confirmatory/ Discovery dataset (Fig. 3a, using 757 out of 1000 features with available data). We classified Cancer Cell Line Encyclopedia (CCLE) mRNA profiles according to subtype derived from CPTAC Confirmatory/Discovery, similarly to that of TCGA pan32 mRNA profiles (Supplementary Fig. 6c). We also classified tumors and cell lines profiled by RPPA according to subtype derived from CPTAC Confirmatory/ Discovery, the tumor dataset from TCGA[5,6] (Fig. 3b) and the cell line dataset from CCLE[33] (Supplementary Fig. 6b). For the RPPA datasets, we used as a classifier the set of represented total protein features from which a significant association with a particular subtype was observable in the CPTAC Confirmatory/Discovery dataset ($p < 0.001$ by $t$-test, based on logged and centered protein expression values).

**Differential expression and pathway analyses.** Differential expression between comparison groups was assessed using $t$-tests on log-transformed values (base 2). False discovery rates (FDRs) were estimated using the method of Storey and Tibshirani[34]. Significantly enriched GO annotation categories were computed using one-sided Fisher's exact tests and SigTerms software[35], based on the entire set of 9764 unique proteins represented in at least three of the five cancer types. Protein interaction network analysis used the entire set of human protein–protein interactions cataloged in Entrez Gene (downloaded June 2017). Entrez protein interactions with yeast two-hybrid experiments providing the only support for the interaction were not included in the analysis. Graphical visualization of networks was generated using Cytoscape[36].

**Gene signature analyses.** We computed gene expression signature scores associated with pathway (e.g., scores for EMT, NRF2/KEAP1, hypoxia, KEGG: Glycolysis/Gluconeogenesis, KEGG: Pentose Phosphate pathway, KEGG: Fatty Acid metabolism, KEGG: TCA Cycle, and KEGG: Oxidative Phosphorylation or OX-PHOS, k-ras, MYC, YAP1, WNT, and NOTCH) as follows. We normalized log base 2-transformed values for proteins in the CPTAC dataset within each cancer type (standard deviations from the median of the given cancer type). For NRF2/ KEAP1, hypoxia, WNT, NOTCH, and KEGG signatures, we computed the average expression of the set of genes within a given signature. For k-ras, MYC, and YAP1 signatures, normalized expression profiles were scored for the above signatures using our $t$-score metric[37]. We generated gene signature scores of NRF2/ KEAP1 pathway as described[38], based on four different signatures[39]. The hypoxia signature was the set of canonical HIF1A targets from Harris[40]. We generated gene transcription signature scores of YAP1 pathway, based on four different signatures[39]. MYC signature (from data by Coller et al. [41]) was from ref. [42], and the Settleman k-ras sensitivity signature was from ref. [43]. WNT signature was taken directly from ref. [44] (summing up values for WNT antagonist, agonist, and target genes). NOTCH signature was taken from ref. [45]. For TCGA pan32 cohort ($n = 10,224$ RNA-seq profiles), gene signature scores associated with the above pathways were also computed as described above, based on transcription data, in a previous study[2]. For visual display, we normalized pathway signature scores across samples to standard deviations from the median.

**Statistical analysis.** All $p$ values were two-sided unless otherwise specified. All tests were performed using log2-transformed expression values. Visualization using heat maps was performed using both JavaTreeview (version 1.1.6r4)[46] and matrix2png (version 1.2.1)[47].

**Reporting summary.** Further information on research design is available in the Nature Research Reporting Summary linked to this article.

## Data availability

All data used in this study are publicly available. The CPTAC datasets (both Confirmatory/Discovery and CPTAC-TCGA) referenced during the study are available from the CPTAC data portal website (https://cptac-data-portal.georgetown.edu/ cptacPublic/). TCGA data RNA-seq data are available through the Genome Data Commons (https://gdc.cancer.gov/) and the Broad Institute's Firehose data portal (https://gdac.broadinstitute.org). The TCGA RPPA dataset is available from the TCPA portal (http://tcpaportal.org/tcpa/). Cancer Cell Line Encyclopedia (CCLE) datasets are available from the CCLE website (http://www.broadinstitute.org/ccle). The source data underlying Figs. 1–7 are provided as a Source Data file. All the other data supporting the findings of this study are available within the article and its supplementary information files and from the corresponding author upon reasonable request. A reporting summary for this article is available as a Supplementary Information file.

## Code availability

R source code written for this study is provided as part of Supplementary Data 6. Example Excel calculations by which the CPTAC-TCGA proteomic profiles were classified according to proteome-based pan-cancer subtype (Fig. 3a) are provided in Supplementary Data 3.

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

## Acknowledgements

This work was supported by National Institutes of Health (NIH) grant P30CA125123 (C.J.C.).

## Author contributions

Conceptualization: C.J.C.; Methodology: C.J.C., F.C.; Investigation: C.J.C., F.C.; Formal Analysis: C.J.C., F.C., D.S.C.; Data Curation: C.J.C., S.V., D.S.C.; Visualization; C.J.C.; Writing: C.J.C., S.V.; Manuscript Review: F.C., D.S.C.; Supervision: C.J.C., S.V.

## Competing interests

The authors declare no competing interests.
