## [Peer Review File · Nature Communications]

Reviewers' comments:

Reviewer #1 (Remarks to the Author):

NCOMMS 19 24367T

Zhang et al. "Pan-cancer molecular subtypes revealed by mass-spectrometry-based proteomic characterization of more than 500 human cancers "

The authors perform a computational biology analyses using mass spec proteomics data deposited by others in TCGA and CPTAC combined with use of deposited mRNA datasets across cancers with a focus on breast, colon, ovarian, renal and uterine cancer) to identify "classes" within and between tumors comprised of mRNA and protein expression signatures. There was the expected amount of agreement between mRNA and protein expression profiles for individual genes (r values of 0.2-0.4). They compare protein expression with the mRNA derived signatures and vice versa. The TCGA and CPTAC proteome data sets were used for training and validation. Overall there were 10 mRNA defined subsets of cancer ("c" subsets) and the proteomics classifiers identified 8 of these, and they likewise have 10 protein ("k" subsets) defined subsets and the mRNA classes mapped to 7 of these. (One of the protein only subsets was only found in renal cancer). They have also provided a "data portal" for others to look at their analyses of the data. No independent experimental validation either of protein expression or function are presented.

Comments to the Authors:

This report represents a detailed computational biology of mRNA and proteomics datasets that have been publically deposited for cancers. As such it is essentially a data resource and in the Results section draws several conclusions from the authors' analyses. At the end of the day, the utility of such studies and this paper will come down to what they can help us understand about cancer biology and pathogenesis, direct us towards new rationale therapeutics, and ultimately provide answers that could benefit patients. With these thoughts in mind, there are several things the authors need to do to make this into a useful, and "understandable" resource for others.

1. The paper is beyond dense in its presentation and writing. I would urge the authors to state each of their major findings, present the key evidence behind these statements and then put all of the detailed discussion in figure legends or supplementary sections. While I thought the figures and one table were good, I repeatedly got bogged down and lost in reading this paper. If I did not have to review it, I would have thrown up my hands and tossed it in the wastebasket at an early stage. In other words, the authors need to make this more "digestable."
2. Do any of the protein classifiers involved proteins that are mutated or aberrantly expressed in the cancers? An obvious question. Whatever the answer is we need to know it.
3. Is there a set of proteins that appear to be useful for immunohistochemical or RPPA studies? While the authors do not have to test these, it would be extremely important to know if there are a subset of proteins that could be identified for each of the classes that could be used in IHC or RPPA studies. These would be ones that could be applied to independently validate and test the different classes.
4. How many of the proteins in each subset are from the tumor and how many from the tumor microenvironment? I realize that some of the sets are entirely microenvironment – but for the others what is known? This also makes the reason for point number 3 above, to be able to use IHC to see which are tumor and which represent the microenvironment.
5. Is there any connection between tumor oncogenotypes and the protein classes found? Again, I don't care what the answer is, we just need to know. It is possible, that the protein classes cut across oncogenotypes and thus provide entirely different sets of information. I would not be surprised if that is the case for something like KRAS mutations -but we need to know.
6. The authors need to provide a "Sweave" <https://stat.ethz.ch/R-manual/R-devel/library/utils/doc/Sweave.pdf> or similar document with this manuscript to allow other computational biologists easily verify the many complex processes and steps the authors have

used.

7. One would want this to be able to help patients. An obvious approach would be to perform proteomics, determine the class the patient's tumor falls into, and from that select the best therapy for that patient. While that is beyond the scope of the current paper, an obvious case would be that a specific signaling pathway(s) are implicated in an individual patient's tumors that could be targeted. One example would be in which patients may be susceptible to checkpoint blockade immunotherapy. While the authors have complex figures depicting pathways, I have no easy way of knowing how this applied to an individual patient's tumor. They could annotate all of the CPTAC and TCGA cases with their classifier and indicate for each the pathways that appear to be active. Maybe there is this somewhere but I couldn't find it.

8. Finally, since patients' tumors will be having mutation analyses, mRNA expression analyses, and say now proteomics analyses, what is the best way to integrate this data to have some kind of combined classifier. These data already exist for TCGA (and probably CPTAC) – so what are the authors' thoughts on how to create an integrated classifier?

9. I went to the cited website and found some basic data analyses possible.

John D. Minna, M.D.

Reviewer #2 (Remarks to the Author):

The presented manuscript by Y. Zhang entitled "Pan-cancer molecular subtypes revealed by mass-spectrometry-based proteomic characterization of more than 500 human cancers" summarizes the effort to analyze and classify different cancer types from The Cancer Genome Atlas (TCGA) and the Clinical Proteomic Tumor Analysis Consortium (CPTAC) at the proteomic level. The authors first highlight that the proteomics data replicates cancer subtype clusters previously established by pan-cancer mRNA analysis. This is followed by classification of the cancer types into subtypes solely on the proteomics level resulting in partially with mRNA subtypes overlapping clusters. Gene ontology enrichment of the clusters then aims to highlight the suspected biological differences between distinct clusters. Overall the manuscript describes a lot of data but fails to distill and present the information. Thus, the novel insights to cancer classification are not immediately obvious. Furthermore, comparing clusters with previously published cancer subtype classifications of the same datasets are missing. In short, the manuscript in its current form cannot be recommended for publication.

Specific comments to the authors (in arbitrary order):

Overall the manuscript is hard to read with long compound sentences. In addition, the structure of the current manuscript does not help as it is very repetitive (a notion underscored by the similarity of figures 1, 2, and 3). Instead of describing in great detail the data from each comparison, it might be better to highlight the most relevant and informative comparisons and relegate the others to the supplementary material. For example, it is not clear how meaningful the comparison with the RPPA data is without a more general discussion about the (dis)similarities of LC/MS- and RPPA-based proteomic analyses.

The manuscript would be greatly helped by an overview scheme that clearly highlights the different sample and data sets and their relationships. It is not always clear how independent/independent those different cohorts are; similarly, the differences/commonalities are not clear, so that it is often difficult to appreciate the results.

Similarly, the manuscript would substantially benefit from more detailed descriptions of the feature selection for the different classifiers (e.g. 198 features in the CPTAC-TCGA dataset vs. 532 features in the CPTAC Confirmatory/Discovery dataset). A Venn-diagram to highlight the overlap of identified proteins in the different datasets with the features from the mRNA classifier would

improve the understanding.

Line 115 and line 121 (examples): "Except for the "c5" class representing basal-like breast cancer, ..." – given that their current figures are hard to decipher it would be really helpful if they clearly indicated with e.g. a differently colored box, what they are referring to.

Figure 1c bottom panel: it is not clear whether/where they ever refer to the bottom panel of Figure 1c, as it is not clear where they are referring to the "1000 class-specific genes".

Line 144: "..., mapping values from the top 100 differential proteins for each subtype (Figure 2b)..." – Figure 2b refers to a 1000 class-specific proteins. Where do the 100 proteins come to play? Please clarify.

The de novo classification at the proteome level is reflected in most clusters already at the mRNA level. While novel clusters 'k2' and 'k3' overlap two distinct mRNA clusters and are a subset of an mRNA cluster, respectively, the manuscript would benefit from detailed comparison and bioinformatic analysis of this finding with the previously published sub-clusters of the CPTAC-TCGA datasets.

Page 11, lines 247-249: The authors describe the use of 162 extracellular matrix proteins used for clustering. However, related Figure 6 as well as Figure 6 legend indicate the use of 606 extracellular matrix proteins. Similar disconnects between numbers mentioned in the text and the figures can be found throughout the text and have to be carefully revised.

Some of the correlation matrices are accompanied by protein lists. However, those lists are rarely discussed in the main text of the manuscript. As such, please either remove those lists, or describe and discuss those proteins lists in more detail.

Figure 3b: the color decoding based on the cancer type is rather useless as it is close to impossible to unambiguously assign the colors of the minuscule rectangles to the font colors of any of the abbreviated cancer types.

Figure 4a: the authors might want to rethink the color coding based on the differential t-statistics as not all readers will be able to make use of/interpret the provided numbers.

The authors compare mRNA gene expression with protein expression. On page 16, line 387, the authors describe the CPTAC Confirmatory/Discovery dataset at the protein level though using the term unique genes as defined by Entrez Identifiers. In the proteogenomic perspective this is not a sound statement as genes can result in multiple isoforms with different protein sequence and function. As the study presented in the manuscript used quantitative information a main concern is how/whether such isoforms were quantitatively collapsed to the gene level. This may affect the overall clustering and Gene Ontology enrichment analysis.

In the methods section the authors describe in multiple places parameter choice and reasoning "consistent with our previous study". This is irritating as for example Supplementary Figure 1a-c indicates that other cluster numbers for the proteomic de novo clusters could have been chosen. The reasoning why the authors chose 10 clusters therefore is missing in this manuscript and only referred to in the previous published work for mRNA clusters.

The authors describe the assignment of subtypes to external cancer cases using correlation with subtype profiles. It appears that all external cases are assigned without mention of a correlation cutoff for assignment or the quality of the assignment.

Page 25, line 570: In Figure 3 legend the authors mention a TCGA Confirmatory/Discovery cohort

that has not been mentioned anywhere in the manuscript. The assumption is that the CPTAC Confirmatory/Discovery cohort is meant here.

Our responses and comments are marked in red italics.

Reviewer #1 comments:

This report represents a detailed computational biology of mRNA and proteomics datasets that have been publically deposited for cancers. As such it is essentially a data resource and in the Results section draws several conclusions from the authors' analyses. At the end of the day, the utility of such studies and this paper will come down to what they can help us understand about cancer biology and pathogenesis, direct us towards new rationale therapeutics, and ultimately provide answers that could benefit patients. With these thoughts in mind, there are several things the authors need to do to make this into a useful, and "understandable" resource for others.

We thank the reviewer for evaluating our work.

1. The paper is beyond dense in its presentation and writing. I would urge the authors to state each of their major findings, present the key evidence behind these statements and then put all of the detailed discussion in figure legends or supplementary sections. While I thought the figures and one table were good, I repeatedly got bogged down and lost in reading this paper. If I did not have to review it, I would have thrown up my hands and tossed it in the wastebasket at an early stage. In other words, the authors need to make this more "digestible."

We deeply thank the reviewer for having the fortitude to patiently work through our manuscript to provide these constructive comments. We have revised and reorganized the manuscript accordingly. The more significant structural change in the revision was our moving the original Figure 1 to Supplementary Materials. This figure examines our previous RNA-seq-based subtypes in the CPTAC proteomic data. On further reflection, using an entire main figure to examine a different set of pan-cancer subtypes that would not be central to the findings of our present study would unnecessarily complicate the main narrative. The new main Figure 1 provides an overview scheme (as requested by Reviewer #2) highlighting the samples and datasets and profiled gene features involved. We use the new Figure 1 as an entry point into the Results section, introducing the datasets used and describing our overall strategy: first using the CPTAC Confirmatory/Discovery cohort to define the proteome-based subtypes and then using the TCGA-related cohorts to independently observe these subtypes. Also, where possible, we have moved statements involving secondary findings or Supplementary Figures from the main text to the Supplementary Figure Legends.

We have also worked to improve the readability of the revised manuscript using the Grammarly.com Premium software. We had used the free version of Grammarly.com for the initial version, which does basic grammar and conciseness checks, but in response to these reviewer comments we splurged for the Premium subscription, which highlighted many issues of clarity, including long and difficult to read sentences, which we then broke up into smaller sentences, as well as our excessively using passive versus active voice.

2. Do any of the protein classifiers involved proteins that are mutated or aberrantly expressed in the cancers? An obvious question. Whatever the answer is we need to know it.

In column BI of the "CPTAC Confirm-Disc TOTAL" Excel tab of the revised Supplementary Data 2 (which provides all protein-level correlations and relevant protein or gene annotation), we indicate

membership of any genes in the COSMIC consensus somatic mutation gene list from Sanger (<https://cancer.sanger.ac.uk/census>, version 90). Of the 12447 proteins in the CPTAC Confirmatory/Discovery datasets, there are 534 genes in the COSMIC list, of which 45 are included in the classifier of 1000 proteins of Figure 2b. One can get these 45 genes by selecting “yes” for both column BI and column BG, using the Excel Data Filters feature. The 45 genes do not represent a statistically significant enrichment for COSMIC genes within the protein classifier, however.

3. Is there a set of proteins that appear to be useful for immunohistochemical or RPPA studies? While the authors do not have to test these, it would be extremely important to know if there are a subset of proteins that could be identified for each of the classes that could be used in IHC or RPPA studies. These would be ones that could be applied to independently validate and test the different classes.

In columns BJ and BK of the “CPTAC Confirm-Disc TOTAL” Excel tab of the revised Supplementary Data 2 (which provides all protein-level correlations and relevant protein or gene annotation), we indicate antibody availability for RPPA for IHC, respectively (referring to the MD Anderson core website or the Human Protein Atlas, respectively). Information provided includes catalog number or HPA accession number. Of the 1000 proteins in the classifier of Figure 2b, 960 would have some antibody availability for IHC according to Human Protein Atlas (one can get these using the Excel Data Filters feature on columns BI and BK).

In Figure 3b of the manuscript (also shown below), we classified the 7694 TCGA cases with RPPA data according to proteome-based pan-cancer subtype. Here we used the expression patterns for a top set of 99 proteins distinguishing between the ten subtypes. For this RPPA analysis we did not limit ourselves to the top 1000 proteins (which represents an arbitrary cutoff), but instead we used as a classifier the set of total protein features from which a significant association with a particular subtype was observable in the CPTAC Confirmatory/Discovery dataset at a significance of $p < 0.001$. Evidence of the presence of specific proteome-based subtypes in the external TCGA cohorts came from observations of subtype calls as made by different data platforms showing significant overlap, e.g., CPTAC-TCGA proteomic subtyping based on mass-spectrometry versus RPPA (Figure 3c, also shown below), and TCGA pan32 subtyping based on mRNA versus RPPA (Figure 3d). Where subtypes did not significantly overlap between data platforms (e.g. k1 and k9 subtypes), we might attribute this to a number of factors. These factors would include differences of the cohort considered from that of CPTAC Confirmatory/Discovery and platform-specific differences (e.g., RPPA platform had few available features specific to k8-k10 subtypes, Figure 3a).

4. How many of the proteins in each subset are from the tumor and how many from the tumor microenvironment? I realize that some of the sets are entirely microenvironment – but for the others what is known? This also makes the reason for point number 3 above, to be able to use IHC to see which are tumor and which represent the microenvironment.

In the revised manuscript, we have carried out an analysis to estimate whether a given protein might be specific to the tumor epithelium or to the surrounding microenvironment. For this we used public mRNA expression profiling data of tumor samples undergoing Laser Capture Microdissection (LCM). Using LCM, both tumor epithelium and tumor stroma components can be profiled separately. In the new Supplementary Figure 4b (also shown below), we take the set of 1000 proteins found to best distinguish between the ten proteome-based subtypes (the proteins from Figure 2b, top 100 over-expressed proteins for each of the 10 subtypes as indicated below), and alongside these differential patterns we show the corresponding differential mRNA patterns comparing tumor epithelium versus tumor stroma, using three independent datasets (breast, colorectal, and prostate). The results indicate that proteins most highly over-expressed in k2, k3, k6, and k7 subtypes tend to represent components of the tumor stroma, while proteins most highly over-expressed in the other subtypes tend to represent the tumor epithelium. This finding aligns with our other findings regarding the predominant influence of the tumor microenvironment within distinct subsets of human cancer, while different microenvironmental components are found to be involved with different tumor subtypes. In columns BL-BN of the “CPTAC Confirm-Disc TOTAL” Excel tab of the revised Supplementary Data 2 (which provides all protein-level

correlations and relevant protein or gene annotations), we provide the corresponding tumor epithelium vs tumor stroma comparisons at the mRNA level for each protein in the CPTAC dataset. This would allow one wishing to do a more in depth analysis of a particular protein to at least have some estimate regarding whether the protein is part of the tumor epithelium or the stroma. An important caveat here is that proteins expressed in both cancer and non-cancer cells may not necessarily be easy to distinguish using this analysis approach.

5. Is there any connection between tumor oncogenotypes and the protein classes found? Again, I don't care what the answer is, we just need to know. It is possible, that the protein classes cut across oncogenotypes and thus provide entirely different sets of information. I would not be surprised if that is the case for something like KRAS mutations -but we need to know.

In the revised manuscript we have added a Supplementary Figure 5 (also shown below), which represents somatic mutations and associated pathways across proteome-based pan-cancer subtypes in the CPTAC-TCGA cohort. (For CPTAC Confirmatory/Discovery cohort, mutation data are still pending for some of the cancer types.) In our previous TCGA RNA-seq subtyping paper (PMID: 29440175), we had annotated all of the TCGA cases according to a curated set of genes involved in oncogenic pathways.

Supplementary Figure 5a shows, for each proteome-based pan-cancer subtype, the significances of enrichment of selected mutation events for each gene within the given subtype versus the rest of the tumors. Supplementary Figure 5b provides a pathway-centric view of nonsilent gene mutations and copy alterations in CPTAC-TCGA cohort (based on n=299 cancer cases with available exome sequencing data, as well as on the pathway groupings represented in part a). Supplementary Figure 5c shows, by pan-cancer subtype, the significances of enrichment of gene alteration events for each pathway within any particular subtype versus the rest of the cases. These results indicate that while a number of pathway-level or individual gene-level DNA alterations surveyed were moderately represented within specific pan-cancer subtypes, no strong connections between tumor oncogenotypes and proteome-based subtypes were evident.

6. The authors need to provide a “Sweave” <https://stat.ethz.ch/R-manual/R-devel/library/utils/doc/Sweave.pdf> or similar document with this manuscript to allow other computational biologists easily verify the many complex processes and steps the authors have used.

In the revised manuscript, we have added a “Code availability” section per Nature Communications guidelines. The R source code written for this study is now provided as part of Supplementary Data 6. Example Excel calculations by which the CPTAC-TCGA proteomic profiles were classified according to proteome-based pan-cancer subtype (Figure 3a) are now provided in Supplementary Data 3 (“classify CPTAC-TCGA by k1-k10” tab). We had used R code to define proteome-based subtypes (using the publicly-available ConsensusClusterPlus R package). R code was also used to generate 1000 random permutations of the subtype assignments for the datasets external to CPTAC Confirmatory/Discovery (whereby in each permutation test, the gene ordering of the external dataset was made random relative to CPTAC subtype classifier, and subtype assignments were made using the “best fit” class with the highest correlation, results presented in Supplementary Figure 7).

Across our entire study, our integrative analysis work required us to use a number of tools in addition to R, including Excel, Adobe Illustrator, Cytoscape, and JavaTreeview/Mat2png, all of which are widely available and accessible. For example, in generating the pathway diagrams, we would find an image off of the internet or a review article that represented the genes and relationships of interest, and then we manually drew a version of the diagram in Illustrator and colored the protein boxes according to differential expression. We would have no code for automatically generating the pathway diagram figures or the Cytoscape networks or the JavaTreeview heat maps. Also, downloading the expression data tables from CPTAC Data Portal and transforming and normalizing the expression values for analysis, as described in the Methods, had to be done manually using Excel. Given all of this, we would not be able to provide a Sweave document that programmatically goes through all of the analyses from scratch and generates figures at a push of a button, but we do wish to ensure that all of our methods are transparent and reproducible.

Per Nature Communications guidelines, we now provide with the manuscript revision a separate source data file, which contains the raw data underlying the main figures or points to where these data reside in Supplementary data. All of the data and protein lists underlying the figures in the manuscript are available in the Supplementary Data, including the normalized expression data matrix for the top differentially expressed proteins, differentially t-statistics according to subtype for all proteins, pathway signature scoring for each cancer case, and GO term associations used to select for protein subsets featured in Figure 5-7. While there is a lot of data and work put into the figures and analyses, it should be clear to a computational researcher what was specifically done. We have the subtype discovery using ConsensusCluterPlus (R code made available), classification of external datasets by CPTAC patterns (R code provided in Supplementary Data 6 and example calculations in Excel provided in Supplementary Data 3), GO term analysis using SigTerms freeware package (PMID: 18812437, Supplementary Data 4), heat maps, pathway diagrams (the underlying t-statistics and p-values and FDR values provided in Supplementary Data 2), and Cytoscape networks (Supplementary Data 5). Importantly, there is no “secret sauce” algorithm or proprietary tool involved here that would be holding back researchers from being able to follow our own analysis work or to carry out similar types of work with other datasets in future studies.

7. One would want this to be able to help patients. An obvious approach would be to perform proteomics, determine the class the patient’s tumor falls into, and from that select the best therapy for that patient. While that is beyond the scope of the current paper, an obvious case would be that a specific signaling pathway(s) are implicated in an individual patient’s tumors that could be targeted. One example would be in which patients may be susceptible to checkpoint blockade immunotherapy. While the authors have complex figures depicting pathways, I have no easy way of knowing how this applied to an individual patient’s tumor. They could annotate all of the CPTAC and TCGA cases with their classifier and indicate for each the pathways that appear to be active. Maybe there is this somewhere but I couldn’t find it.

Supplementary Data File 1 provides patient-level subtyping and molecular features as an Excel file. Information provided for each patient includes assigned subtypes (e.g. CPTAC proteome-based subtype and TCGA mRNA-based subtype), pathway signature scoring, and immune signature scoring. Results are provided for the CPTAC Confirmation/Discovery cohort, the CPTAC-TCGA cohort, the TCGA pan32 mRNA/RPPA cohort. One can also look up differential statistics for a given protein using Supplementary Data File 2, or differential expression patterns for a given protein (e.g. according to subtype, stage/grade, patient characteristics, etc.) can be visualized using the provided UALCAN Data Portal. One limitation with associating elevation of pathway signature or markers with therapeutic response is that we do not necessarily know the sensitivity threshold expression levels associated with a given protein or proteins. From Supplementary Data 3, PDI is actually not represented in CPTAC data (an apparent limitation of the mass spectrometry proteomics platform), and PDL1/CD274 (detected only for Renal and Uterine cancer types) does not appear differentially expressed in any of the subtypes, though UALCAN Data Portal, into which we incorporated the CPTAC data, might reveal other associations of interest.

8. Finally, since patients’ tumors will be having mutation analyses, mRNA expression analyses, and say now proteomics analyses, what is the best way to integrate this data to have some kind of combined classifier. These data already exist for TCGA (and probably CPTAC) – so what are the authors’ thoughts on how to create an integrated classifier?

In the new Supplementary Figure 4d (also shown below), we provide a Cluster-of-clusters analysis (COCA, PMID:25109877), as a first attempt to integrate subtype classifications of the 364 TCGA-CPTAC cases, at the levels of proteome (present study, assignments of proteome-based subtype to TCGA-CPTAC cases represented in main Figure 3a), transcriptome (from PMID: 29440175), and DNA-level pathway alterations (curated for TCGA cases in PMID: 29440175). In the clustered data matrix, red denotes membership for the given cancer case in the given subtype. We see here that DNA-based classifications cluster separately from the proteome-based on transcriptome-based classifications, and that there is some observed overlap between the proteome-based and transcriptome-based classifications (e.g. mRNA.c5/protein.k4, mRNA.c6/protein.k5, mRNA.c7/protein.k6, mRNA.c8/protein.k7).

Future studies could presumably find better ways to integrate RNA and protein information for use as a combined classifier. In terms how our study result might apply to subtyping patient tumors in the clinical setting, it seems more likely that a select set of protein markers of potential therapeutic relevance underlying just one or more subtypes of interest might be examined in tumors, e.g. using IHC. It seems less likely that a protein-based test to assign each patient tumor to one of our ten proteome-based subtypes would find widespread use in the clinic.

9. I went to the cited website and found some basic data analyses possible.

We are happy that the reviewer could access the UALCAN data portal and perform some basic analyses. This would provide one level of access to these CPTAC proteomic results by the general biomedical research community. Our original paper on TCGA RNA-seq data in UALCAN, published in 2017 (PMID: 28732212), has already garnered over 280 citations to date, and our data portal web site gets an average of 500 visits in a single day from researchers across the globe. By making CPTAC data a part of UALCAN we can make the data available in an easily understandable and analyzable format to researchers who do not have extensive computational skills. In addition, the supplementary Excel data files provided as part our present study compile all relevant protein-level statistics and information, as

well as cancer case-level information according to assigned subtype and associated pathways. Researchers may use Excel to look up any proteins of individual interest, as well as access the data and protein sets underlying the figures in the paper.

Reviewer #2 comments:

The presented manuscript by Y. Zhang entitled “Pan-cancer molecular subtypes revealed by mass-spectrometry-based proteomic characterization of more than 500 human cancers” summarizes the effort to analyze and classify different cancer types from The Cancer Genome Atlas (TCGA) and the Clinical Proteomic Tumor Analysis Consortium (CPTAC) at the proteomic level. The authors first highlight the that the proteomics data replicates cancer subtype clusters previously established by pan-cancer mRNA analysis. This is followed by classification of the cancer types into subtypes solely on the proteomics level resulting in partially with mRNA subtypes overlapping clusters. Gene ontology enrichment of the clusters then aims to highlight the suspected biological differences between distinct clusters.

Overall the manuscript describes a lot of data but fails to distill and present the information. Thus, the novel insights to cancer classification are not immediately obvious. Furthermore, comparing clusters with previously published cancer subtype classifications of the same datasets are missing. In short, the manuscript in its current form cannot be recommended for publication.

We thank the reviewer for carefully reading our work. We address the above points raised in the comments below.

1. Overall the manuscript is hard to read with long compound sentences. In addition, the structure of the current manuscript does not help as it is very repetitive (a notion underscored by the similarity of figures 1, 2, and 3). Instead of describing in great detail the data from each comparison, it might be better to highlight the most relevant and informative comparisons and relegate the others to the supplementary material. For example, it is not clear how meaningful the comparison with the RPPA data is without a more general discussion about the (dis)similarities of LC/MS- and RPPA-based proteomic analyses.

We have revised and reorganized the manuscript in light of these comments. The more significant structural change in the revision was our moving to Supplemental the original Figure 1, which figure examines our previous RNA-seq-based subtypes in the CPTAC proteomic data. On further reflection, using an entire main figure to examine a different set of pan-cancer subtypes that would not be central to the findings of our present study would unnecessarily complicate the main narrative. The new Figure 1 provides the requested overview scheme (comment #2) highlighting the samples and datasets and profiled gene features involved. We use the new Figure 1 as an entry point into the Results section, introducing the datasets used and describing our overall strategy: first using the CPTAC Confirmatory/Discovery cohort to define the proteome-based subtypes and then using the TCGA-related cohorts to independently observe these subtypes.

We have worked to improve the readability of the revised manuscript using the Grammarly.com Premium software. We had used the free version of Grammarly.com for the initial version, which does basic grammar and conciseness checks, but in response to these reviewer comments we splurged for the

Premium subscription, which highlighted many issues of clarity, including long sentences (as noted by the reviewer), which we then broke up into smaller sentences.

In the revised manuscript we have added a reference to our previous review on the RPPA data platform (PMID: 26185419), which discusses the differences between RPPA and mass-spectrometry-based proteomics. We acknowledge in the manuscript that factors involved in any disparate results between RPPA and mass spec could include platform-specific differences. At the same time, we do see significant overlaps in subtyping assignments between CPTAC-TCGA versus RPPA (Figure 3c), and between TCGA pan32 mRNA versus RPPA (Figure 3d), which suggests real biology being reflected in each respective data platform. Based on Reviewer #1's comments, there should be a fair amount interest from some readers in how results from mass spectrometry might be translated to the RPPA platform.

2. The manuscript would be greatly helped by an overview scheme that clearly highlights the different sample and data sets and their relationships. It is not always clear how independent/independent those different cohorts are; similarly, the differences/commonalities are not clear, so that it is often difficult to appreciate the results.

The new Figure 1a provides the requested overview scheme (also shown below) highlighting the samples and datasets involved.

3. Similarly, the manuscript would substantially benefit from more detailed descriptions of the feature selection for the different classifiers (e.g. 198 features in the CPTAC-TCGA dataset vs. 532 features in the CPTAC Confirmatory/Discovery dataset). A Venn-diagram to highlight the overlap of identified proteins in the different datasets with the features from the mRNA classifier would improve the understanding.

The new Figure 1b (also shown below) provides a Venn diagram with the numbers of shared gene features (protein or mRNA levels) among CPTAC Confirmatory/Discovery, TCGA RPPA, and TCGA pan32 mRNA datasets. As noted in our response to Reviewer comment #12, CPTAC proteomic and TCGA

transcriptomic data, as provided by their respective public data portals, were processed at the gene level, rather than at the protein isoform or mRNA transcript levels.

The old Figure 1, which has the 198 vs 532 protein features noted by the Reviewer, has been moved to the Supplementary Materials as Supplementary Figure 1, in the interests of simplifying the main narrative. As noted in the Supplementary Figure 1 legend of the revision, the set of 198 genes for CPTAC-TCGA is based on available protein data on the original set of 854 genes, taken from the set of 5863 proteins represented in all three cancer types. For CPTAC Confirmatory/Discovery dataset, the set of 532 genes is taken from the set of 12247 proteins found for any of the five cancer types represented in that dataset. The above gene lists are accessible from the Supplementary Data File 3, which provides protein-level information for CPTAC-TCGA and CPTAC Confirmatory/Discovery datasets. Using the Excel data filters for column X of the “CPTAC-TCGA” tab gives us the 198 genes, and column BO of the “CPTAC Confirm-Disc TOTAL” tab gives us the 532 genes.

4. Line 115 and line 121 (examples): “Except for the “c5” class representing basal-like breast cancer, ...” – given that their current figures are hard to decipher it would be really helpful if they clearly indicated with e.g. a differently colored box, what they are referring to.

Regarding the old Figure 1 referred to by the above, we have moved this Figure to Supplemental in the manuscript revision, in the interests of having a more stream-lined main narrative. The “c5” pan-cancer class and its basal-like association were described in our previous mRNA-focused study (PMID: 29440175). Regarding the proteome-based k4 subtype, which also represents basal-like breast cancer, we demonstrate the association in Supplementary Figure 4a (also shown below). In Supplementary Figure 4a, expression patterns of the PAM50 gene set (38 genes represented in CPTAC datasets) in the mRNA profile dataset from Hoadley et al. and CPTAC Confirmatory/Discovery proteomic dataset are represented, with the Hoadley basal-like breast cancers and CPTAC k4 tumors being highlighted.

5. Figure 1c bottom panel: it is not clear whether/where they ever refer to the bottom panel of Figure 1c, as it is not clear where they are referring to the “1000 class-specific genes”.

As noted above, we have moved Figure 1 from the previous version into the supplemental of the revised version, in order to simplify the main narrative. The legend of the new Supplementary Figure 1c notes the following: “The bottom heat map shows differential expression patterns for a set of 1000 proteins in CPTAC Confirmation/Discovery found to best distinguish between the ten classes (see Methods).” As this bottom panel heat map simply illustrates the widespread protein expression differences observed among the transcriptome-based pan-cancer classes, but would not necessarily be germane to the proteome-based subtype differences, we do not refer to this particular result in detail in the main text.

6. Line 144: “..., mapping values from the top 100 differential proteins for each subtype (Figure 2b)...” – Figure 2b refers to a 1000 class-specific proteins. Where do the 100 proteins come to play? Please clarify.

There are ten proteome-based subtypes, and so taking the top 100 over-expressed proteins for each subtype gives us 10x100=1000 proteins in total. In the manuscript revision, we have clarified this point in the Figure 2b legend and associated main text.

7. The de novo classification at the proteome level is reflected in most clusters already at the mRNA level. While novel clusters ‘k2’ and ‘k3’ overlap two distinct mRNA clusters and are a subset of an mRNA cluster, respectively, the manuscript would benefit from detailed comparison and bioinformatic analysis of this finding with the previously published sub-clusters of the CPTAC-TCGA datasets.

In the new Supplementary Figure 4c (also shown below), we show the significances of overlap between the proteome-based pan-cancer subtype assignments made for CPTAC-TCGA cases in the present study (rows), with molecular-based subtype assignments (columns) made previously for a subset of cases in CPTAC- or TCGA-led studies.

Overall, the proteome-based pan-cancer molecular subtypes show significant concordances with other molecular subtype designations. The two immune-related subtypes appear distinct from each other in terms of associations with the previous subtypes (e.g. k2 but not k3 associating with Ovarian Immunoreactive subtype), and the two stroma-related subtypes also appear distinct from each other in terms of associations with the previous subtypes (e.g. k6 but not k7 associating with Ovarian Mesenchymal subtype, and k7 associating the Ovarian Stromal subtype).

8. Page 11, lines 247-249: The authors describe the use of 162 extracellular matrix proteins used for clustering. However, related Figure 6 as well as Figure 6 legend indicate the use of 606 extracellular matrix proteins. Similar disconnects between numbers mentioned in the text and the figures can be found throughout the text and have to be carefully revised.

The 162 number should be 606 (typo, carry over from related text used to describe Figure 5a results). Thanks so much for catching this.

9. Some of the correlation matrices are accompanied by protein lists. However, those lists are rarely discussed in the main text of the manuscript. As such, please either remove those lists, or describe and discuss those proteins lists in more detail.

In the revised manuscript, we include a parenthetical statement to Figure 2b legend: "...lists provide examples of differentially expressed proteins but these would not necessarily have more importance over the other proteins in the heat map, full lists provided in Supplementary Data 2 and 3." Also, as noted in Figure 3b legend, proteins highlighted by name in that figure were individually significantly associated with the given subtype (at the $p < 0.001$ level) in TCGA RPPA dataset. The old Figure 1 with its protein list has been moved to the Supplementary section in the manuscript revision. As noted above, the protein lists in the main figures are intended to provide examples to the reader of what types of proteins may be associated with which subtypes. We wanted to take advantage of space within the main figures to provide examples, even if we cannot list all of the top differential proteins by name. A trained eye can note, for example, immune-related proteins associated with k2 and k3, as well as other potentially interesting proteins. The lists simply represent a starting point for interested readers to hopefully dig through the supplementary data files 2 and 3, which provide the complete lists of proteins represented in the heat

maps. Given the stated concerns by the reviewers regarding the density of results already the main text, we will forgo discussing these example proteins in detail within the main text itself.

10. Figure 3b: the color decoding based on the cancer type is rather useless as it is close to impossible to unambiguously assign the colors of the minuscule rectangles to the font colors of any of the abbreviated cancer types.

In the revised manuscript, we have added a note to the Figure 3 legend: “Patient-level subtyping and cancer type information for all datasets represented are provided in Supplementary Data 1.” It is true that with the large numbers of cases of the 32 cancer types involved here, it may not always be possible for one to readily discern every cancer type associated with a pan-cancer subtype. The main message here regarding the tumor color bar is that the subtypes as called in TCGA RPPA dataset do in fact span multiple tissue-based cancer types, beyond the six cancer types of the CPTAC dataset.

11. Figure 4a: the authors might want to rethink the color coding based on the differential t-statistics as not all readers will be able to make use of/interpret the provided numbers.

In the revised manuscript, we have added a note to the legends for Figures 4, 5, and 6 which use the purple-cyan t-statistic heat map representation: “(bright purple/cyan, highly significant; black, not significant; shades close to black, borderline significant)”. As with the expression heat maps, the t-statistic heat maps are intended to give the general reader a qualitative assessment of the significance of the changes, without one getting lost in the numbers and p-values. Basically, if one can see at least some color, then the association would be at least borderline significant. The actual p-values associated with the protein-level t-statistics are provided in supplementary data 3.

12. The authors compare mRNA gene expression with protein expression. On page 16, line 387, the authors describe the CPTAC Confirmatory/Discovery dataset at the protein level though using the term unique genes as defined by Entrez Identifiers. In the proteogenomic perspective this is not a sound statement as genes can result in multiple isoforms with different protein sequence and function. As the study presented in the manuscript used quantitative information a main concern is how/whether such isoforms were quantitatively collapsed to the gene level. This may affect the overall clustering and Gene Ontology enrichment analysis.

Actually, CPTAC proteomic data, as provided by the CPTAC Data Portal (<https://cptac-data-portal.georgetown.edu/cptacPublic/>) and associated CPTAC-led publications, were processed at the gene level or Entrez ID level, rather than at the protein isoform level. Our study follows the convention of previous CPTAC-led studies, whereby different isoforms for the same protein were not separately considered in the present study, as a simplification. Therefore, for the overall clustering and Gene Ontology enrichment analyses in our present study, there was no need for us to “pick” which protein isoform we wanted to include for a particular gene. In principle, it should be possible to process the raw mass spectrometry data from CPTAC into protein isoforms, but such a version of the processed data are apparently not provided by CPTAC. In the revised manuscript, we have made reference to the above in the new Figure 1 legend and in the Methods.

13. In the methods section the authors describe in multiple places parameter choice and reasoning “consistent with our previous study”. This is irritating as for example Supplementary Figure 1a-c

indicates that other cluster numbers for the proteomic de novo clusters could have been chosen. The reasoning why the authors chose 10 clusters therefore is missing in this manuscript and only referred to in the previous published work for mRNA clusters.

In the revised Supplementary Figure 2 legend (formerly Supplementary Figure 1), and in the Methods section under “Pan-cancer molecular subtype discovery,” we include additional notes regarding our selection of the 10-cluster solution. We note that, based on the metrics provided in Supplementary Figure 2, additional clustering solutions would more subtypes may potentially be discoverable, although these additional subtypes would involve fewer numbers of cases and would represent less of the global variation. While additional molecular subtypes could be defined arbitrarily (e.g. by increasing the ConsensusClusterPlus k), such subtypes would also need to have some unique biology associations for them to be relevant.

As the Reviewer indicates, we are not able to definitely state that the 10-subtype solution is the only solution that may be applied to human cancers. In our study, we wanted to make sure that we were going deep enough that our previously identified mRNA-based subtypes (PMID: 29440175) would have the opportunity to be found within the proteome-based subtypes if they were in fact present. As noted in Supplementary Figure 2c legend, the k1 subtype (metabolism-related and analogous to the “c1” mRNA-based subtype) appears at the k=9 solution and up. Every molecular subtyping study must utilize its own analytic approach and decision points, while here we did explore various options and found the final solution presented to represent a good framework for the exploration of differences between cancer subtypes. In the Discussion section and elsewhere, we leave the door open for future studies involving larger numbers of cancer cases or different cancer types to identify additional pan-cancer subtypes.

14. The authors describe the assignment of subtypes to external cancer cases using correlation with subtype profiles. It appears that all external cases are assigned without mention of a correlation cutoff for assignment or the quality of the assignment.

We used permutation testing to assess the overall strength of the correlations behind the CPTAC proteome-based subtype assignments made to external datasets. These results are provided as Supplementary Figure 7, the first two panels of which (for TCGA RPPA and TCGA-CPTAC datasets) are shown below.

In each permutation test, the gene ordering of the given external dataset was made random relative to CPTAC subtype classifier, and subtype assignments were made using the “best fit” class with the highest

correlation. The distribution of the average best fit subtype correlations from each of the permuted datasets are shown in each figure panel (representing 1000 best subtype similarity scores), along with the average best fit correlations for each of the ten pan-cancer classes in the actual, non-permuted datasets. We see that the actual average similarity score for each pan-cancer subtype is much greater than what a distribution based on a random scenario would allow.

In addition, Supplementary Data File 3 of the revised manuscript provides example calculations in Excel, by which the CPTAC-TCGA proteomic profiles were classified according to proteome-based pan-cancer subtype (in the “classify CPTAC-TCGA by k1-k10” tab, assigned subtype is computed in row 1, referring to calculations in rows 1005-1016, results used for Figure 3a). The Pearson’s correlations underlying the assigned subtype had $p < 0.03$ for all 364 cases, and all but seven cases had a Pearson’s correlation with $p < 0.001$.

15. Page 25, line 570: In Figure 3 legend the authors mention a TCGA Confirmatory/Discovery cohort that has not been mentioned anywhere in the manuscript. The assumption is that the CPTAC Confirmatory/Discovery cohort is meant here.

We have corrected this typo in the revision, thanks for catching this.

REVIEWERS' COMMENTS:

Reviewer #1 (Remarks to the Author):

The authors have responded appropriately to all of the reviewers' comments including significant editing and re-organization of their manuscript.

Reviewer #2 (Remarks to the Author):

Overall, the manuscript by Chen et al. entitled "Pan-cancer molecular subtypes revealed by mass-spectrometry-based proteomic characterization of more than 500 human cancers" has improved. The authors moved Figure 1 into the supplementary section and provide an overview figures of the data instead, which is exceedingly helpful in guiding the reader to understand the relationships/overlaps between the different datasets. While this and breaking up some of the longer sentences help with the flow of the manuscript, the manuscript still contains too much data, which 'hide' the relevant information and result, i.e. it is still an almost futile exercise to follow the analyses and to understand the messages and conclusions. Such shortcoming cannot be corrected by software such as Grammarly, as the content needs major revision, not only the grammar.

Concepts and protein sets are still referred to in much detail even during later use (see "Tumor stroma-related differences..." section at lines 237 and 239). While the overall section speaks about the 606 extracellular matrix-related proteins, the authors draw a comparison with the associations resulting from "the 162 proteins" that were the focus of the previous section and are immune-related.

Furthermore, the discussion section of the manuscript highlights the fact that many of the findings of the authors have been made previously and the overall result of the manuscript merely provides a resource for future larger scale studies. Instead of focusing on one or two truly novel findings, the reader is confronted with three rather similar highly complex figures, which are almost impossible to comprehend.

Given the missing novelty, the lack of focus on the relevant and novel findings, and the difficulties comprehending the current manuscript, I do not believe the manuscript warrants publication in Nature Communications.

Our responses and comments are marked in red italics.

Reviewer #1 comments:

The authors have responded appropriately to all of the reviewers' comments including significant editing and re-organization of their manuscript.

We thank the reviewer for evaluating our work.

Reviewer #2 comments:

Overall, the manuscript by Chen et al. entitled “Pan-cancer molecular subtypes revealed by mass-spectrometry-based proteomic characterization of more than 500 human cancers” has improved. The authors moved Figure 1 into the supplementary section and provide an overview figures of the data instead, which is exceedingly helpful in guiding the reader to understand the relationships/overlaps between the different datasets. While this and breaking up some of the longer sentences help with the flow of the manuscript, the manuscript still contains too much data, which ‘hide’ the relevant information and result, i.e. it is still an almost futile exercise to follow the analyses and to understand the messages and conclusions. Such shortcoming cannot be corrected by software such as Grammarly, as the content needs major revision, not only the grammar.

The editors have provided guidance to address the concerns regarding the readability of the manuscript. While they do not believe that a significant re-write is necessary at this stage, they suggested a series of edits to help improve the clarity of the work.

The editors recommended tailing the end of each results section with a brief summary of the key take-home messages. In the revised manuscript, we now end each results section with a brief summary of the key take-home messages. An example summary for the immune-related section (Figure 5) is as follows: “In summary, regarding immune system-related differences, k2 subtype involved the adaptive immune response and T-cell activation, and k3 subtype involved the humoral immune response. Proteins that distinguish k2 from k3 subtypes include markers of T-cells (high in k2) and markers of mast cells, neutrophils, or macrophages (all high in k3), as well as complement system pathway proteins (high in k3). These distinctions between k2 and k3 were not evident in previous mRNA-based subtyping², where k2 and k3 tumors associated together as a single group.”

In order to help provide a sharper focus on the unique features of each group, in the brief summary paragraphs, we note the unique features of particular interest distinguishing the subtypes. An example for this regarding the immune-related subtypes is above. Another example regarding the stroma-related subtypes is as follows: “Protein markers that distinguish k6 from k7 subtypes include: FN1, IGFBP3, ITGAV, LOX, LOXL2, MMP11, MMP13, MMP14, and THBS1 (all high in k6, Figure 6a); and collagen VI and associated proteins (high in k7, Figure 6c).”

Concepts and protein sets are still referred to in much detail even during later use (see “Tumor stroma-related differences...” section at lines 237 and 239). While the overall section speaks about the 606 extracellular matrix-related proteins, the authors draw a comparison with the associations resulting from “the 162 proteins” that were the focus of the previous section and are immune-related.

The tumor stroma section should just refer to the 606 protein of Figure 6a. It should not be referring to the 162 proteins of the previous immune-related section. In this latest revision, we have checked for this.

Furthermore, the discussion section of the manuscript highlights the fact that many of the findings of the authors have been made previously and the overall result of the manuscript merely provides a resource for future larger scale studies. Instead of focusing on one or two truly novel findings, the reader is confronted with three rather similar highly complex figures, which are almost impossible to comprehend.

In response to these and similar comments, the editors recommended that we provide a more explicit honest discussion of the benefits of this approach, such as what does proteomic analysis provide that other methodologies do not (e.g. referring to our k8-10 groups which were not reflected in the transcriptome). Also, we should discuss how this work could be taken further.

At the end of Discussion section, we have a final paragraph that broadly summarizes our study and its key implications, including implications involving more attention by the research community being focused on pathways that have received less study in the context of cancer: “In summary, our study has uncovered proteome-based pan-cancer subtypes on the basis of mass-spectrometry-based proteomics, which platform offers far more protein features over other proteomic platforms such as RPPA⁵. Where specific proteome-based subtypes are found reflected in subtypes defined on the basis of mRNA patterns², the existence of such subtypes in human cancer has even stronger support with these independent observations involving the proteome in addition to the transcriptome. On the other hand, our study also uncovers cancer subtypes not found in previous transcriptome-based studies. Observations specific to the proteome include two different immune system-related subtypes (reflected as a common subtype in the transcriptome), one involving the adaptive immune response and the other involving the humoral immune response. The pathway associations according to proteome-based subtype may help in drawing attention to pathways and processes, previously examined in the laboratory setting, which are shown here to be involved in appreciable subsets of human cancer. In addition to the more heavily-studied pathways such as cancer metabolism and T cell signaling, other pathways and processes to be considered further in the context of cancer would include those involving complement activation, collagen VI, Golgi apparatus, hemoglobin complex, and endoplasmic reticulum.” We also now note in the introduction section that the previous CPTAC-led marker studies were each focused upon a single cancer type, while here we are bringing all these data together in pan-cancer analyses.

Given the missing novelty, the lack of focus on the relevant and novel findings, and the difficulties comprehending the current manuscript, I do not believe the manuscript warrants publication in Nature Communications.

Our key findings are summarized throughout the manuscript, including main Table 1 and the added summary paragraphs after each Results section and the summary in Discussion section.

Our study may actually be the first CPTAC pan-cancer study of this nature. The mass-spectrometry-based proteomics platform of CPTAC offers far more protein features over other proteomic platforms such as RPPA. Therefore, this gives us an opportunity to look at pan-cancer subtypes in a way that has not been done before. Where specific proteome-based subtypes are found reflected in subtypes defined on the basis of mRNA patterns, this would still represent a very important finding, as the existence of such subtypes in human cancer now has even stronger support with these independent observations involving the proteome

in addition to the transcriptome. At the same time, our study also uncovers cancer subtypes not found in previous transcriptome-based studies. Observations specific to the proteome include two different immune system-related subtypes (reflected as a common subtype in the transcriptome), one involving the adaptive immune response and the other involving the humoral immune response. The k8, k9, and k10 proteome-based subtypes are also unique to our study.

We do not believe that our manuscript and figures would be difficult for most readers to comprehend, at a level beyond what is typically shown in TCGA-led or CPTAC-led studies. Both the Creighton group and TCGA Network have recently published a number of pan-cancer studies involving 10,000 tumors and multiple data platforms. All of these studies involve a new approach or question applied to these highly complex datasets. As with our other studies, we start here with an inherently complex dataset and systematically arrive at some broad conclusions, while showing the data and relevant patterns in the interests of full transparency. Using all of these CPTAC data from different cancer types, we can broadly categorize these into ten major groups, which is a powerful statement.